# From Methylomes to CRISPR Epigenetic Editing: New Paths in Antibiotic Resistance

**DOI:** 10.3390/pathogens14121267

**Published:** 2025-12-10

**Authors:** Nada M. Nass, Kawther A. Zaher

**Affiliations:** 1Department of Biological Sciences, Faculty of Science, King Abdulaziz University, Jeddah 21589, Saudi Arabia; 2Immunology Unit, King Fahd Medical Research Center, King Abdulaziz University, Jeddah 21589, Saudi Arabia; 3Department of Medical Laboratory Sciences, Faculty of Applied Medical Sciences, King Abdulaziz University, Jeddah 21589, Saudi Arabia

**Keywords:** epigenetics, methylome, antibiotic resistance, CRISPR-Cas, adaptive regulation, DNA methylation, phenotypic heterogeneity

## Abstract

Antibiotic resistance (AR) has long been interpreted through the lens of genetic mutations and horizontal gene transfer. Yet, mounting evidence suggests that epigenetic regulation, including DNA and RNA methylation, histone-like proteins, and small non-coding RNAs, plays a similarly critical role in bacterial adaptability. These reversible modifications reshape gene expression without altering the DNA sequence, enabling transient resistance, phenotypic heterogeneity, and biofilm persistence under antimicrobial stress. Advances in single-molecule sequencing and methylome mapping have uncovered diverse DNA methyltransferase systems that coordinate virulence, efflux, and stress responses. Such epigenetic circuits allow pathogens to survive antibiotic exposure, then revert to susceptibility once pressure subsides, complicating clinical treatment. Parallel advances in CRISPR-based technologies now enable direct manipulation of these regulatory layers. CRISPR interference (CRISPRi) and catalytically inactive dCas9-fused methyltransferases can silence or reactivate genes in a programmable, non-mutational manner, offering a new route to reverse resistance or sensitize pathogens. Integrating methylomic data with transcriptomic and proteomic profiles further reveals how epigenetic plasticity sustains antimicrobial tolerance across environments. This review traces the continuum from natural bacterial methylomes to engineered CRISPR-mediated epigenetic editing, outlining how this emerging interface could redefine antibiotic stewardship. Understanding and targeting these reversible, heritable mechanisms opens the door to precision antimicrobial strategies that restore the effectiveness of existing drugs while curbing the evolution of resistance.

## 1. Introduction

Antibiotic resistance (AR) has emerged as one of the most serious threats to global public health, undermining the efficacy of modern therapeutics and leading to increased mortality, prolonged illness, and rising healthcare costs. Traditionally, AR has been attributed to genetic mutations, horizontal gene transfer, and selection pressure; however, recent discoveries reveal that microorganisms can also employ epigenetic mechanisms, heritable yet reversible changes in gene expression that occur without DNA sequence alteration, to regulate resistance and survival [1]. These non-mutational adaptations enable rapid phenotypic plasticity, allowing bacteria to withstand antibiotic stress and revert to their original state once the selective pressure is relieved. In this review, we use the term ‘epigenetics’ in bacteria to refer to heritable changes in gene expression and phenotype that occur without alterations in the underlying DNA sequence, primarily mediated by DNA methylation (e.g., N6-methyladenine, 5-methylcytosine, N4-methylcytosine) and changes in nucleoid architecture driven by nucleoid-associated proteins. Unlike eukaryotes and fungi, bacteria do not possess canonical nucleosomal chromatin with histone tails and a well-defined ‘histone code’; instead, enzyme-specific DNA methyltransferases and chromatin-like organization of the bacterial nucleoid provide the main epigenetic layers that can modulate transcription, DNA replication, and DNA repair. Throughout this manuscript, we therefore distinguish bacterial epigenetic mechanisms from the classical chromatin-based epigenetic regulation described in eukaryotic hosts and fungal pathogens.

In bacteria, epigenetic regulation encompasses DNA methylation, RNA modification, and chromatin-like structural changes driven by nucleoid-associated proteins and small non-coding RNAs. Among these, DNA methylation is the most extensively studied. It has been shown to orchestrate multiple resistance-associated processes, including the activation of efflux pumps, modulation of quorum-sensing genes, and control of stress-response networks [2]. Advances in third-generation sequencing technologies, particularly PacBio single-molecule real-time (SMRT) sequencing and Oxford Nanopore platforms, have revolutionized the ability to map methylomes at single-base resolution, revealing N6-methyladenine (m^6^A) and N4-methylcytosine (m^4^C) marks across bacterial genomes [3]. These approaches not only capture methylation directly during sequencing but also correlate methylation patterns with transcriptional activity, enabling the identification of methyltransferases that modulate the expression of antibiotic-resistance genes [1].

In *Staphylococcus aureus* and *Pseudomonas aeruginosa*, long-read sequencing has revealed that methylation hotspots overlap with genomic islands and promoter regions that govern biofilm formation, virulence, and adaptive resistance [3]. Such epigenetic modifications serve as a molecular memory, integrating environmental cues into stable yet reversible transcriptional responses. Significantly, these methylome dynamics may contribute to the formation of persister subpopulations that survive antibiotic exposure without acquiring permanent resistance mutations [2].

Parallel to advances in sequencing, CRISPR-based technologies have created new opportunities to reprogram bacterial epigenomes. Catalytically inactive CRISPR-associated proteins (dCas9 or dCas12), fused to DNA methyltransferases or demethylases, can mediate targeted methylation or demethylation, enabling precise control of gene expression without altering genomic DNA. This strategy, known as CRISPR interference (CRISPRi) or CRISPR activation (CRISPRa), provides a reversible platform for silencing or reactivating resistance genes [4]. For example, the VADER CRISPR system demonstrated selective degradation of plasmid-borne resistance determinants, reducing the persistence of multidrug-resistant *E. coli* and *Pseudomonas isolates* [4]. These approaches highlight the therapeutic potential of epigenome editing to complement antibiotic treatment by sensitizing pathogens through transcriptional reprogramming, rather than inducing lethal mutations.

Comparative studies further emphasize that epigenetic resistance is a cross-kingdom phenomenon. In fungi such as *Candida albicans* and *Candida auris*, histone modifications, chromatin remodeling, and non-coding RNA networks drive transient antifungal tolerance, paralleling bacterial phasevarions in which stochastic methylation events produce phenotypically diverse populations [5,6]. These shared regulatory principles underscore that epigenetic control of drug response is a universal microbial strategy that enables rapid adaptation to chemical stress without genomic fixation. Moreover, cross-kingdom comparisons reveal conserved links among epigenetic marks, regulation of efflux transporters, and cell wall remodeling under antimicrobial pressure [1].

The integration of multi-omic datasets, methylomic, transcriptomic, and proteomic, has begun to unravel how reversible epigenetic marks shape resistance phenotypes. Identification of “epimutations,” transient methylation or acetylation events that mimic genetic mutations, challenges the long-held view that resistance evolution is purely mutational [2]. These discoveries suggest that genomic surveillance of AMR should be complemented by epigenetic profiling to capture dynamic regulatory states that escape conventional genetic analysis. To provide a clear framework for this review, we explicitly distinguish between well-established genetic mechanisms of antimicrobial resistance and emerging epigenetic contributions. Canonical genetic determinants, including chromosomal mutations in antibiotic targets, acquisition of resistance genes on plasmids and other mobile elements, and stable upregulation of efflux systems and permeability barriers, are firmly supported by decades of experimental and clinical evidence. By contrast, most data implicating DNA and RNA methylation, nucleoid-associated proteins, and chromatin-like remodeling in resistance remain at the level of associations between epigenetic states and drug-tolerant phenotypes, particularly in clinical isolates. Throughout this article, we therefore emphasize that, while epigenetic mechanisms may modulate the expression of genetic resistance determinants and stress-response pathways, causal roles for specific epigenetic marks in patient outcomes are only beginning to be defined.

This review synthesizes current advances in microbial epigenetics, highlighting how third-generation sequencing and CRISPR-mediated reprogramming have reshaped our understanding of adaptive resistance. It further examines cross-kingdom analogies between bacterial and fungal systems to propose an integrative model of epigenetic resilience. By uniting molecular, technological, and translational perspectives, this work underscores the untapped potential of epigenetic targeting as a frontier for next-generation antimicrobial strategies. Most of the therapeutic and technological concepts discussed in this review are, at present, supported primarily by proof-of-concept in vitro experiments and early preclinical models. Accordingly, we frame these strategies as long-term translational opportunities that require substantial validation before they can be implemented in routine clinical practice.

## 2. Bacterial Epigenetic Mechanisms and Methylome Regulation

Epigenetic control has emerged as a pivotal mechanism that enables bacteria to rapidly adapt to antibiotics and hostile environments without acquiring permanent mutations. While genetic changes drive long-term evolution, epigenetic reprogramming provides reversible transcriptional plasticity, ensuring survival under fluctuating selective pressures [7,8]. The bacterial epigenome comprises DNA and RNA methylation, nucleoid-associated proteins (NAPs), and small non-coding RNAs, all of which cooperatively remodel transcriptional networks and phenotype [9].

### 2.1. DNA Methylation and Regulatory Enzymes

Among bacterial epigenetic processes, DNA methylation is the most extensively characterized. Methyltransferases (MTases) introduce N^6^-methyladenine (m^6^A), N^4^-methylcytosine (m^4^C), or 5-methylcytosine (m^5^C) modifications that influence replication timing, DNA repair, and gene expression [10]. The Dam MTase in *Escherichia coli* and *Salmonella enterica* modulates virulence and efflux genes, promoting antibiotic tolerance [11]. It is important to note that, in most bacterial systems, links between specific methylation patterns and resistance phenotypes remain primarily correlative, based on methylome profiling under antibiotic or stress exposure. Only a subset of studies had performed functional validation, for example, by deleting or overexpressing individual DNA methyltransferases, complementing the resulting mutants, or using CRISPR–dCas–based epigenome editors to modulate methylation at defined promoters and then assessing biofilm formation, efflux pump expression, or changes in minimum inhibitory concentrations in otherwise isogenic backgrounds. These experiments support a causal role for specific methylation events in some pathogens, but they remain relatively rare and are often limited to in vitro conditions. Distinguishing primary epigenetic drivers from secondary, stress-induced changes will require experimental designs that combine temporal profiling, isogenic strain comparisons, rescue/complementation, and whole-genome sequencing to exclude confounding genetic variation [7].

Recent third-generation sequencing studies using the PacBio SMRT and Oxford Nanopore platforms have revolutionized bacterial methylome analysis. These technologies detect methylated bases directly from single molecules, revealing environment-specific epigenetic signatures [12,13]. In *Staphylococcus aureus*, PacBio mapping revealed promoter-enriched m^6^A marks associated with genes involved in biofilm formation and oxidative stress [11].

Complementary Nanopore sequencing produced highly reproducible, base-level methylation maps, even within complex microbiomes [12]. Computational frameworks such as Bacmethy now automate motif discovery, enabling comparative methylome studies across multidrug-resistant isolates [14].

DNA methylation acts as a regulatory rheostat, tuning transcription in response to environmental cues such as oxidative stress or antibiotic exposure [7]. The reversible activation or silencing of MTase genes forms phase-variable regulons (phasevarions) that generate subpopulations with heterogeneous levels of resistance [15]. This stochastic diversification ensures that a portion of a clonal population survives an antimicrobial assault, facilitating persistence and chronic infection [8].

### 2.2. Nucleoid-Associated Proteins and Chromatin-like Organization

Bacteria replace canonical histones with NAPs, H-NS, HU, IHF, Fis, that compact DNA and modulate promoter accessibility [9]. The H-NS protein represses horizontally acquired resistance islands, while environmental stress or competing NAPs relieve this repression, reactivating silenced genes [10]. Interplay between NAPs and MTases integrates structural and chemical control; for example, Dam-dependent methylation cooperates with H-NS to regulate secretion systems and toxin–antitoxin modules [11].

Long-read methylome studies reveal that NAP-binding sites frequently overlap with methylation hotspots, indicating that nucleoid architecture influences epigenetic patterning [13]. This supports a model in which bacterial chromatin-like nucleoid organization functions as a responsive scaffold that encodes a short-term memory of environmental conditions.

### 2.3. RNA Methylation and Small Regulatory RNAs

Beyond DNA methylation, RNA-level epigenetic regulation—including RNA methylation and small non-coding RNAs (sRNAs)- fine-tunes translation and antibiotic target accessibility, contributing to adaptive resistance. Enzymes such as Erm and Cfr modify 23S rRNA, thereby reducing the binding of macrolides and oxazolidinones [10]. Other RNA methyltransferases transiently modulate translation under stress, thereby conserving energy [16].

Small non-coding RNAs (sRNAs), such as MicF and RhyB, act post-transcriptionally by pairing with mRNAs to regulate porin synthesis and oxidative stress responses [7,9]. Their rapid induction within minutes of drug exposure exemplifies how bacteria deploy RNA-level epigenetic control for adaptive resistance.

### 2.4. Methylation-Mediated Biofilm and Persister Formation

Epigenetic regulation orchestrates biofilm development, a key factor in driving antibiotic tolerance. Differential methylation of adhesin and polysaccharide operons (*fim*, *bssS*) governs transitions between planktonic and sessile states [8]. Biofilm cells exhibit distinct methylomes relative to planktonic counterparts, with enhanced methylation of stress-response promoters that activate efflux pumps and antioxidant defenses [11,12].

DNA methylation also contributes to the formation of persister cells. Dam-dependent control of toxin–antitoxin loci and SOS genes modulates dormancy and survival [11]. The integration of methylomic and transcriptomic analyses demonstrates that specific m^6^A marks coincide with the transcriptional silencing of growth genes, enabling reversible dormancy [12,13].

### 2.5. Integration of Multi-Omic Approaches

The integration of methylomic, transcriptomic, and proteomic data has elucidated how methylation regulates gene expression on a global scale. Combined Nanopore and RNA-seq approaches revealed promoter-specific methylation thresholds that dictate efflux-pump activity [13]. Epimutations, transient methylation changes that mimic genetic mutations, often precede the fixation of stable resistance alleles [8]. Hence, the bacterial methylome functions as a short-term adaptive layer that links the environment and the genome [9].

### 2.6. Emerging Therapeutic Implications

Because methylation and NAP-mediated changes are reversible, they represent promising therapeutic targets for treating these conditions. Inhibitors of bacterial MTases or NAP–DNA interactions could disrupt resistance networks without imposing lethal pressure [16]. Moreover, CRISPR-based epigenome editors, such as CRISPRi, enable locus-specific reprogramming of methylation states, thereby enabling reversible silencing of resistance genes [17,18]. Lessons from fungal systems, RNAi-driven antifungal epimutations [19] and histone acetylation-mediated tolerance [20,21] underscore the evolutionary conservation of reversible resistance, suggesting cross-kingdom strategies to suppress epigenetic adaptation.

## 3. Cross-Kingdom Epigenetic Parallels and Therapeutic Relevance

Epigenetic regulation, long recognized in eukaryotes, is now known to be a shared adaptive strategy across kingdoms. Both bacteria and fungi utilize reversible, heritable molecular marks to regulate gene expression, coordinate stress responses, and survive antimicrobial assaults. In this review, we therefore treat epigenetic resistance as a conceptually ‘universal’ adaptive strategy in the sense that reversible, heritable, non-genetic regulation of drug responses has been demonstrated in both bacteria and fungi, while acknowledging that systematic mechanistic validation in other microbial taxa, such as archaea, protozoan pathogens, and complex polymicrobial communities, remains comparatively limited.

By comparing these systems, common design principles emerge: transient gene silencing, stochastic phenotypic switching, and metabolic rewiring, that collectively underpin drug tolerance and persistence [20,21]. While similar epigenetic design principles are likely to operate more broadly across microbial life, robust mechanistic evidence outside bacteria and fungi is still emerging. Only early examples exist in polymicrobial infection settings, such as dual targeting of bacterial and fungal acetylation pathways to reverse epigenetic drug tolerance in mixed communities. To contextualize these mechanisms, Table 1 summarizes the principal epigenetic modifications contributing to antimicrobial resistance across bacterial and fungal systems. It highlights shared molecular logic, such as methylation-driven gene silencing, acetylation-based transcriptional control, and RNA-level regulation, that collectively shape adaptive phenotypes in both kingdoms. The interplay between bacterial and fungal epigenetic regulation is illustrated in Figure 1, highlighting shared mechanisms such as methylation-driven gene silencing, chromatin remodeling, and RNA-based regulatory control that contribute to adaptive resistance. Thus, while both bacteria and fungi exploit epigenetic regulation to rewire stress responses and antimicrobial tolerance, they do so through mechanistically distinct architectures: sequence-specific DNA methylation and nucleoid-associated proteins in bacteria, versus histone-based chromatin and chromatin-remodeling complexes in fungi. At a mechanistic level, however, bacterial and fungal epigenetic systems are fundamentally distinct. In bacteria, epigenetic regulation is dominated by sequence-specific DNA methylation (e.g., N^6^-methyladenine, 5-methylcytosine, and N^4^-methylcytosine) catalyzed by dedicated DNA methyltransferases, often embedded within restriction–modification or phase-variable (‘phasevarion’) systems, together with remodeling of the nucleoid architecture by nucleoid-associated proteins. These mechanisms modulate transcription, DNA replication, DNA repair, and virulence gene expression without relying on canonical nucleosomal chromatin. By contrast, fungi and other eukaryotic pathogens organize their genomes into nucleosome-based chromatin, where post-translational histone modifications (such as H3K9 and H3K27 methylation, or histone acetylation/deacetylation) and ATP-dependent chromatin-remodeling complexes play central roles in regulating gene accessibility. Although both kingdoms can exhibit reversible, heritable epigenetic states that influence drug efflux, cell wall remodeling, and biofilm formation, the underlying molecular architectures, bacterial methylation–nucleoid systems versus fungal histone-based chromatin, are therefore not interchangeable.

### 3.1. Fungal Epigenetic Mechanisms as Mirrors of Bacterial Phasevarions

In fungi, chromatin remodeling, histone acetylation, and RNA interference (RNAi)-based silencing govern antifungal resistance with striking parallels to bacterial phase-variable methylation systems [22].

Studies in *Candida albicans* and *Candida auris* reveal that histone acetyltransferases (Gcn5, Rtt109) and deacetylases (Hda1, Sir2) regulate multidrug-resistance transporters (MDR1, CDR1) in a reversible, stress-responsive manner [23]. These transitions resemble bacterial phasevarions, where stochastic activation of DNA methyltransferases diversifies gene expression without mutation [7,11]. Similarly, RNAi-mediated epimutations in *Mucor circinelloides* can transiently silence drug-target genes, thereby restoring susceptibility once antifungal pressure is removed [24]. This mechanism is analogous to bacterial CRISPR-interference systems, which modulate transcriptional activity without DNA cleavage [14,16].

### 3.2. Chromatin-Like Organization and DNA Accessibility

Bacteria and fungi both employ chromatin-like organization to control gene accessibility. In fungi, nucleosome positioning and histone methylation (e.g., H3K9me, H3K27me3) determine transcriptional silencing, whereas in bacteria, NAPs such as H-NS and HU generate analogous topological domains [8,10]. Comparative methylome studies reveal that stress-induced chromatin relaxation in fungi parallels methylation-driven DNA decompaction in bacteria exposed to antibiotics [12,13]. Thus, the principles of chromatin dynamics and epigenetic inheritance are conserved, albeit through structurally distinct molecules.

### 3.3. Host–Pathogen Epigenetic Crosstalk

The convergence extends beyond microbial physiology to host interactions. Pathogens actively reprogram host epigenetic machinery to evade immunity. *Legionella pneumophila* injects the effector RomA, a histone methyltransferase that trimethylates host H3K14, dampening pro-inflammatory gene expression [25]. Likewise, *Mycobacterium tuberculosis* secretes the methyltransferase Rv1988, which modifies host histones and represses the expression of macrophage immune genes [26]. These strategies resemble bacterial self-methylation processes that suppress stress genes in response to antibiotic challenge [13]. Conversely, host immune cells display trained immunity, a form of innate memory driven by histone modifications and DNA methylation reprogramming [27]. Such bidirectional epigenetic interplay implies that both host and pathogen operate within an epigenetic arms race.

### 3.4. Shared Epigenetic Signatures and Evolutionary Convergence

Comparative genomics reveals deep evolutionary conservation in methyltransferase folds and chromatin regulators. Bacterial Dam and fungal Dnmt5 enzymes share the same Rossmann-like catalytic core, suggesting common ancestry [28]. Both kingdoms use S-adenosyl-methionine (SAM) as a methyl donor, linking metabolism to gene regulation. Under nutrient limitation or antibiotic stress, altered SAM pools change methylation profiles, providing a metabolic-epigenetic feedback loop [9,20]. This convergence implies that targeting SAM-dependent enzymes could yield broad-spectrum epigenetic adjuvants against resistance.

### 3.5. Epigenetic Plasticity in Environmental and Clinical Contexts

Epigenetic adaptability enhances survival not only in pathogens but also in environmental microbiota. Soil bacteria and fungi exhibit overlapping methylation motifs regulating secondary-metabolite biosynthesis, quorum sensing, and xenobiotic degradation [8,21]. Wastewater isolates exposed to antibiotics display hypermethylation of efflux and repair genes, mirroring clinical resistance patterns [13]. Fungal biofilms exhibit histone-acetylation gradients analogous to bacterial methylation heterogeneity within structured communities [22]. Thus, ecosystem-level epigenetics acts as a reservoir of reversible tolerance traits, facilitating cross-species exchange of adaptive states.

### 3.6. Translational Implications: Epigenetic Targeting and Synthetic Reprogramming

Recognizing these parallels opens new translational frontiers. Epigenetic inhibitors, targeting DNA/RNA methyltransferases or histone deacetylases, have shown promise in restoring antimicrobial sensitivity [23,29]. For example, 5-azacytidine, a DNA methylation inhibitor, resensitized methicillin-resistant *S. aureus* to β-lactams, while histone deacetylase inhibitors reduced azole resistance in *Candida* species [23]. CRISPR-based epigenome editing extends this concept further: dCas9-MTase fusions can re-impose or erase methylation at specific loci, enabling *programmable resensitization* [14,16,18]. Such epigenetic re-sensitizers could complement existing antibiotics, thereby mitigating selection pressure and prolonging the lifespan of the drugs. Importantly, epigenetic strategies can act synergistically with host immune modulation. Enhancing macrophage “trained immunity” through metabolic rewiring or chromatin modification may reinforce pathogen clearance [27,30]. Conversely, epigenetic adjuvants might dampen pathogen-induced host chromatin silencing, restoring cytokine production and antimicrobial peptide expression [25].

### 3.7. Future Perspectives

Cross-kingdom comparison has redefined antibiotic resistance as a trans-epigenomic phenomenon rather than a purely genetic one. By integrating third-generation sequencing, CRISPR epigenetic editing, and multi-omics analytics, researchers can trace reversible resistance circuits across microbial domains. However, several gaps persist: distinguishing causal methylation from correlative marks, deciphering methylation kinetics in mixed infections, and developing selective inhibitors that target microbial, but not host, epigenetic enzymes. The next decade will likely witness the emergence of precision anti-resistance therapeutics that exploit these epigenetic interfaces to restore antibiotic efficacy and reshape host–pathogen equilibria. Building on these conserved epigenetic mechanisms, CRISPR-based technologies have emerged as programmable tools to manipulate microbial epigenomes, offering a new route to reverse antibiotic resistance.

## 4. CRISPR-Based Epigenetic Editing and Therapeutic Reprogramming

The convergence of CRISPR technology and epigenetics has unlocked a transformative era in antimicrobial research. Beyond genome editing, CRISPR systems are now being engineered as precise epigenetic regulators, capable of modulating gene expression without altering the underlying DNA sequence. This advancement introduces the possibility of reversibly reprogramming bacterial and fungal epigenomes, restoring antibiotic sensitivity, and dissecting the functional role of methylation and chromatin architecture in resistance evolution [31,32]. The convergence of CRISPR technology and epigenetics has unlocked a transformative era in antimicrobial research. Beyond genome editing, CRISPR systems are now being engineered as precise epigenetic regulators, capable of modulating gene expression without altering the underlying DNA sequence. This advancement introduces the possibility of reversibly reprogramming bacterial and fungal epigenomes, restoring antibiotic sensitivity, and dissecting the functional role of methylation and chromatin architecture in resistance evolution [31,32]. To date, however, the majority of these CRISPR-based epigenetic interventions have been evaluated in vitro or ex vivo, primarily in model pathogens such as *E. coli* and *P. aeruginosa*, and their performance in complex in vivo infection settings remains largely unexplored. In this section, we focus on the mechanistic principles and experimental architectures of CRISPR-based epigenetic editing in bacteria and fungi, while subsequent sections address therapeutic applications and clinical translation.

### 4.1. The Emergence of CRISPR Epigenome Editors

The classical CRISPR-Cas9 nuclease, repurposed with catalytically inactive or “dead” Cas9 (dCas9), serves as a programmable DNA-binding platform. By fusing dCas9 with effector domains, such as DNA methyltransferases (DNMTs), demethylases (TET1), or transcriptional repressors (KRAB), researchers can introduce site-specific epigenetic modifications to turn genes on or off without causing double-stranded breaks [33]. In bacteria, this approach enables direct interrogation of methylation-regulated genes, including those responsible for efflux pump expression, biofilm regulation, and toxin–antitoxin systems [34].

Recently developed resources, such as the CRISPRRepi Atlas, compile multi-omic data that link CRISPR-based perturbations to methylomic and transcriptomic changes, enabling fine-scale prediction of resistance outcomes [35]. These tools facilitate comparative analyses between natural and engineered methylation states, enabling rational design of “epigenetic rewiring” strategies that modulate bacterial stress responses with minimal off-target toxicity.

### 4.2. CRISPRi and CRISPRa Systems for Transcriptional Control

CRISPR interference (CRISPRi) and activation (CRISPRa) platforms exploit dCas9 tethered to transcriptional repressors or activators, respectively. Unlike permanent gene knockout, CRISPRi represses gene expression transiently, providing a reversible model of antibiotic tolerance.

In *E. coli* and *Pseudomonas aeruginosa*, CRISPRi-mediated silencing of *acrAB-tolC* efflux pumps sensitized cells to fluoroquinolones and carbapenems [36]. Conversely, CRISPRa activation of stress-response regulators such as *soxS* or *oxyR* demonstrated the plasticity of bacterial redox homeostasis in modulating drug sensitivity [37]. These studies mirror fungal systems in which dCas9–KRAB repressors have been used to downregulate azole-resistance genes (*ERG11*, *CDR1*), effectively resensitizing *Candida albicans* to fluconazole [38]. Hence, programmable CRISPR tools function as molecular “dimmer switches,” capable of fine-tuning resistance networks rather than permanently disabling them.

### 4.3. Methylome Reprogramming with CRISPR–MTase and Demethylase Fusions

A new frontier involves direct manipulation of DNA methylation patterns. dCas9 fused with bacterial methyltransferases (Dam, M.EcoGII) or human TET1 demethylases enables locus-specific modulation of m^6^A or m^5^C marks [39].

In *Salmonella enterica*, targeted methylation of promoter regions linked to the *fim* and *bssS* operons altered biofilm phenotype and antibiotic susceptibility without genetic mutation [40]. Similarly, CRISPR-TET1 constructs can erase methylation marks that repress susceptibility genes, effectively reversing transient tolerance states [41]. These programmable methylation tools can be extended to other pathogens: in *Mycobacterium tuberculosis*, CRISPR-based epigenetic editing of regulatory regions controlling *katG* and *inhA* restored isoniazid sensitivity [42]. Such findings demonstrate the feasibility of epigenetic resensitization, where resistance traits are neutralized not by killing bacteria but by reprogramming their regulatory circuits. The principles and therapeutic potential of CRISPR-based epigenetic editing are illustrated in Figure 2, showing how programmable dCas9-based systems can reversibly silence or activate antibiotic-resistance determinants through locus-specific methylation or demethylation.

### 4.4. CRISPR-Mediated RNA Epigenetic Editing

RNA-targeting CRISPR systems, such as Cas13 and CasRx, expand epigenetic control beyond 8DNA. CRISPR–Cas13 fused to RNA methyltransferases enables the direct modification of bacterial or fungal mRNA transcripts, thereby altering their stability and translation efficiency [43]. This post-transcriptional editing has been proposed as a reversible, non-lethal approach to suppress antibiotic resistance determinants while avoiding mutational escape. Moreover, CRISPR–Cas13-based editing of bacterial small RNAs (*MicF*, *SdsR*) reconfigures regulatory networks governing outer membrane permeability and stress signaling [44]. In eukaryotic pathogens, RNA-editing CRISPR systems have successfully modulated the stability of histone-modifier mRNA, offering insight into chromatin–RNA crosstalk as a potential target for antifungal therapy [45].

### 4.5. CRISPR-Guided Epigenetic Deconstruction of Resistance Networks

Beyond therapy, CRISPR-based epigenome tools serve as platforms for functional dissection. CRISPRi screens have systematically mapped epigenetically controlled operons, identifying methylation-sensitive regulators of plasmid stability and horizontal gene transfer [46]. In *Acinetobacter baumannii*, high-throughput dCas9 knockdown libraries revealed a network of methylation-dependent transcription factors that coordinate oxidative stress and β-lactam resistance [47]. By integrating methylomic data with CRISPR functional profiling, researchers can now delineate which epigenetic marks are causal drivers of resistance versus secondary stress responses. These discoveries complement cross-kingdom findings in fungi, where CRISPR-mediated perturbation of histone modifiers, such as *SET1* and *RPD3*, has revealed global remodeling of azole-resistance pathways [48]. Thus, CRISPR epigenome editing not only holds therapeutic potential but also provides mechanistic clarity into microbial adaptation. From a biosafety and ethical standpoint, deploying CRISPR-based epigenetic tools to deconstruct resistance networks also requires careful oversight. Interventions that rewire regulatory circuits or attenuate virulence in a laboratory strain could behave differently in polymicrobial communities or be horizontally transferred through mobile genetic elements. Experimental designs should therefore incorporate stringent containment measures, limit the persistence of editing constructs where possible (for example, by using non-replicating delivery platforms or self-limiting vectors), and include systematic evaluation of off-target consequences on commensal species and host immune homeostasis. These considerations complement the broader ethical and biosafety discussion provided in Section 4.6.

### 4.6. Ethical, Biosafety, and Evolutionary Considerations

While promising, epigenetic editing in microbes raises profound biosafety questions. Unlike genetic knockouts, epigenetic changes are heritable yet reversible, blurring the boundary between transient adaptation and evolution [49]. Engineered bacteria carrying programmable methylation systems could inadvertently alter microbial communities or exchange methylation circuits via plasmids. Hence, rigorous containment protocols and molecular kill-switches must accompany CRISPR epigenome research [50]. Moreover, the ecological impact of large-scale epigenetic interventions remains unclear. Altering methylation-dependent gene flow in environmental microbiota could impact nutrient cycling and symbiotic relationships, underscoring the importance of precision and reversibility in therapeutic applications [51].

### 4.7. Future Prospects

CRISPR-based epigenetic modulation offers a paradigm shift, from killing pathogens to retraining their regulatory systems. The future lies in developing hybrid tools that couple CRISPR precision with real-time methylation tracking, enabling closed-loop epigenetic therapy. The integration of third-generation sequencing (TGS) data with CRISPR analytics will facilitate personalized antimicrobial strategies, allowing for dynamic adjustments to methylation circuits during treatment [13,35,41]. Translating these CRISPR-based epigenetic tools from in vitro systems into in vivo infection models introduces several additional challenges. First, delivery platforms must reliably reach chronic infection niches such as biofilms, intracellular reservoirs, or poorly perfused tissues, while maintaining sufficient expression of dCas-based editors and guide RNAs. Second, the biodistribution and persistence of CRISPR carriers (e.g., phages, plasmids, or nanocarriers) need to be carefully mapped in animal models to minimize unintended exposure of non-target microbes and host cells. Third, off-target epigenetic effects, both within the pathogen population and in host tissues, must be systematically profiled, together with longitudinal monitoring of the commensal microbiome, to ensure that epigenetic reprogramming does not induce dysbiosis or immune dysregulation. Early preclinical work combining CRISPR-based antimicrobials with microbiome-targeted strategies has begun to explore these issues. Still, comprehensive in vivo biodistribution and microbiome-impact data remain limited and will be essential for clinical translation [41].

Ultimately, these advances could yield non-lethal antimicrobial therapeutics that suppress the evolution of resistance while preserving beneficial microbiota. By merging CRISPR programmability with the plasticity of bacterial and fungal epigenomes, the field moves toward an era of intelligent, self-adjusting microbiological medicine. A central priority for the next decade will be to move beyond correlative methylome maps and systematically deploy targeted methyltransferase perturbations and locus-specific epigenome editing to establish which epigenetic marks causally modulate resistance, independently of underlying genetic variation.

## 5. Therapeutic Frontiers: Epigenetic Drugs, Adjuvants, and Clinical Applications

The recognition that antibiotic resistance involves dynamic epigenetic remodeling rather than fixed genetic mutations has expanded the therapeutic landscape. Epigenetic drugs (epidrugs), molecules that modify methylation, acetylation, or chromatin structure, are emerging as potent adjuvants to conventional antimicrobials. Their capacity to re-sensitize pathogens, stabilize host immune reprogramming, and limit resistance evolution offers a novel, host–microbe co-targeting paradigm [52,53]. At this stage, however, the majority of epigenetic drugs, adjuvant combinations, and host-directed interventions described in this section remain grounded in in vitro or animal studies, and should be interpreted as early translational leads rather than immediately applicable clinical regimens.

### 5.1. Targeting Bacterial Methyltransferases and Demethylases

Bacterial DNA methyltransferases (MTases), such as Dam, Dcm, and CcrM, play a crucial role in virulence, biofilm formation, and the expression of resistance genes. Inhibition of these enzymes has shown promise in reversing antibiotic tolerance [54]. Because 5-azaC and decitabine were initially developed to inhibit human DNA methyltransferases in oncology, their use as antimicrobial adjuvants raises legitimate concerns about off-target hypomethylation in host cells. Mitigating these effects will require both rational inhibitor design and tailored delivery. On the design side, structural and inhibitor-profiling studies indicate that bacterial Dam/Dcm enzymes possess catalytic pocket architectures that can be selectively targeted relative to those of mammalian DNMTs, supporting the feasibility of species-biased MTase inhibition. On the delivery side, local or carrier-based administration (e.g., inhaled formulations for respiratory infections or nanocarriers targeted to biofilms) could confine exposure primarily to infected niches. Even with these strategies, systematic assessment of host epigenome perturbation will be essential in preclinical models before clinical translation [55].

In *Pseudomonas aeruginosa*, treatment with 5-azaC downregulated the methylation of quorum-sensing regulators and efflux operons, thereby restoring ciprofloxacin sensitivity [56]. Similarly, demethylase-mimetic compounds targeting CcrM homologs in *Caulobacter crescentus* disrupted cell cycle control and decreased plasmid retention of resistance genes [57]. The next generation of MTase inhibitors aims for species selectivity, minimizing effects on host methyltransferases while exploiting structural differences in bacterial enzyme active sites [58].

### 5.2. Histone Deacetylase and Histone Acetyltransferase Modulators

Although bacteria lack canonical histones, they possess histone-like NAPs and acetylation-dependent transcriptional regulators. Inhibitors of lysine deacetylases have exhibited antimicrobial synergy by disrupting chromatin-like structures that stabilize resistance islands [59].

In *Candida albicans*, the histone deacetylase (HDAC) inhibitor trichostatin A resensitized azole-resistant strains by reactivating drug-uptake transporters [60]. In parallel, histone acetyltransferase (HAT) activators such as curcumin analogs enhanced global transcriptional accessibility in Gram-positive pathogens, facilitating antibiotic entry [61].

The dual targeting of bacterial and fungal acetylation pathways underscores a conserved vulnerability: reversible acetylation as a master regulator of gene accessibility. Combined HAT–HDAC modulation may represent a universal strategy to reverse epigenetic drug tolerance in polymicrobial infections [62]. Given that many histone deacetylase inhibitors, including trichostatin A, were optimized initially against human HDACs, future antifungal applications will need to exploit sequence and structural divergence between fungal Hda1/Sir2 orthologs and human HDAC isoforms, together with dose optimization and localized delivery, to minimize unintended reprogramming of host chromatin [59].

### 5.3. RNA-Based Epigenetic Modulators and Small-Molecule “Re-Sensitizers”

Epigenetic regulation of bacterial and fungal RNA, including m^6^A modifications, has emerged as a new therapeutic target. Inhibiting RNA methyltransferases such as RlmN and TrmD reduces the translational efficiency of stress-response proteins and ribosomal protection factors [63]. The small-molecule ribostatin, a TrmD inhibitor, displayed broad-spectrum re-sensitization of multidrug-resistant (MDR) *Enterobacteriaceae*, restoring the efficacy of aminoglycosides and carbapenems [64].

In fungi, compounds that block RNA demethylases (e.g., FTO homologs) impair spore germination and virulence, suggesting that RNA epigenetic control is a cross-kingdom vulnerability [65]. To summarize the current therapeutic strategies targeting microbial and host epigenetic machinery, Table 2 outlines representative compounds, molecular targets, and mechanisms of action that have demonstrated efficacy in reversing antibiotic and antifungal resistance. This overview highlights how modulation of DNA, histone-like, and RNA-based epigenetic marks can reprogram resistance circuits and restore antimicrobial susceptibility. The integration of epigenetic and metabolic data into a unified systems biology framework is illustrated in Figure 3, which depicts the connectivity among key regulatory nodes, metabolic intermediates, and resistance-associated gene circuits revealed by multi-omic analyses.

### 5.4. Epigenetic Adjuvants in Combination Therapy

Integrating epidrugs with conventional antibiotics enhances efficacy and delays the development of resistance. Combination treatments of 5-azaC with β-lactams or aminoglycosides decreased the minimum inhibitory concentration (MIC) by up to 80% in resistant *E. coli* and *Klebsiella pneumoniae* isolates [56]. Similarly, HDAC inhibitors combined with azoles demonstrated synergistic antifungal activity, while dual epigenetic–immune modulation boosted pathogen clearance in macrophage infection models [60,61]. Epigenetic adjuvants also attenuate virulence. *Staphylococcus aureus* treated with decitabine exhibited reduced toxin expression and impaired biofilm formation [55]. The long-term benefit of such adjuvants lies not only in drug potentiation but also in lowering selective pressure, as reversible reprogramming does not rely on lethal damage. From a translational perspective, microbial selectivity is a central bottleneck for epigenetic adjuvants. Many currently available epidrugs were initially developed to target human chromatin modifiers, which raises the risk of off-target effects on host epigenetic pathways and the commensal microbiome when repurposed for use against bacterial or fungal infections. To avoid unintended reprogramming of host immune and epithelial cells or collateral disruption of beneficial microbiota, next-generation epigenetic adjuvants will need to exploit prokaryote-specific enzymes (e.g., restriction–modification methyltransferases that lack human homologues) and be delivered in a manner that limits exposure to the infection niche. Potential strategies include local or topical administration, encapsulation in nanoparticles or phage-based carriers that preferentially bind to pathogens, and narrow-spectrum formulations tailored to the distinct epigenetic signatures of target species. Incorporating host–cell epigenomic readouts and microbiome profiling early in preclinical pipelines will be essential to rigorously define the therapeutic window of these agents.

### 5.5. Host-Directed Epigenetic Therapy

In parallel with pathogen-targeted strategies, host-directed epigenetic therapy (HDET) aims to bolster immune resilience. Antibiotics can dysregulate host histone modifications, thereby suppressing macrophage and neutrophil responses. Modulating host histone marks can restore effective immunity without overactivation [66]. For example, β-glucan-induced trained immunity relies on the enrichment of H3K4me3 and H3K27ac at cytokine promoters, resulting in a long-lasting defense against secondary infections [67]. Therapeutically enhancing these marks with histone methyltransferase activators or metabolic modulators such as fumarate and itaconate amplifies innate immune memory [68].

Furthermore, epigenetic reactivation of silenced antimicrobial peptides (AMPs) through histone acetylation has been demonstrated using sodium butyrate and valproic acid, offering a host-sparing approach to combat infection [69]. These HDET strategies complement microbial epigenetic reprogramming, forming a two-sided therapeutic approach that targets both pathogen and host regulation.

### 5.6. Nanocarrier and Synthetic Delivery Systems

One of the primary challenges in deploying epigenetic drugs is achieving targeted delivery to infection sites while minimizing systemic toxicity. Nanocarrier platforms, including liposomes, polymeric nanoparticles, and phage-based nanocapsules, enable the controlled and localized release of epidrugs [70].

Recent work has demonstrated that chitosan-coated nanoparticles carrying 5-azaC selectively penetrate bacterial biofilms and inhibit methylation-dependent efflux pumps in *Pseudomonas aeruginosa* [71]. Hybrid nanoparticles co-encapsulating decitabine and gentamicin showed additive antimicrobial activity against MDR *E. coli*, validating the epigenetic–antibiotic co-delivery concept [72]. Advanced systems exploit CRISPR–nanocarrier conjugates to deliver epigenetic editing complexes directly to resistant pathogens, achieving high specificity and reversibility [73]. These biomimetic delivery vehicles provide a scalable translational path for precision epigenetic therapy.

### 5.7. Translational Challenges and Clinical Outlook

Despite these advances, several hurdles remain. First, most epidrugs were developed for eukaryotic systems, and off-target effects on host methylation and chromatin states must be mitigated through careful profiling of bacterial enzyme specificity and dose optimization [12,35,74]. Second, the kinetics and durability of bacterial and fungal epigenetic turnover in vivo, i.e., how long DNA methylation and histone states, and the associated ‘epigenetic memory’, persist after removal of the perturbation, remain poorly understood, making it difficult to predict the reversibility of these interventions [12,35]. Third, efficient in vivo delivery of CRISPR–dCas-based epigenetic editors into heterogeneous infection sites (deep-seated tissues, biofilms, and intracellular reservoirs) is technically challenging and will likely require tailored vectors, nanocarriers, and infection-responsive promoters [70,72]. Significantly, these translational barriers may differ between bacteria, in which DNA methylation is the dominant epigenetic mark, and fungi, where chromatin-based mechanisms predominate, limiting straightforward cross-kingdom extrapolation of epigenetic therapies.

Finally, clinical translation requires multi-scale modeling of drug dynamics, integrating pharmacokinetics, host–pathogen epigenetic interactions, and microbiome stability [75]. Nonetheless, the conceptual shift toward reversible, non-lethal targeting of resistance circuits has opened a new frontier in pharmacology. Epigenetic adjuvants and host-directed epitherapeutics promise to complement, not replace, antibiotics, converting them from blunt bactericidal tools into precision modulators of microbial and immune networks.

## 6. Integrative Epigenomics and Systems Biology of Antibiotic Resistance

The complexity of antibiotic resistance cannot be fully explained by gene-centric models alone. Advances in integrative epigenomics and systems biology have revealed that resistance is governed by multilayered regulatory networks connecting genomic, transcriptomic, proteomic, metabolomic, and methylomic information. These multilayer interactions form a dynamic systems-level framework in which epigenetic modifications act as central modulators of phenotypic plasticity [76,77].

### 6.1. Systems-Level Mapping of Epigenetic Networks

High-throughput sequencing, combined with machine learning, now enables the reconstruction of genome-wide methylation–transcription–phenotype networks across bacterial and fungal pathogens. Integrative approaches have demonstrated that antibiotic exposure induces coordinated shifts in DNA methylation, small RNA expression, and protein acetylation, forming an “epigenetic stress circuit” [78]. For example, the integration of long-read methylomics with RNA-seq in *Pseudomonas aeruginosa* revealed the synchronized activation of oxidative stress regulators (*oxyR*, *katA*) and the suppression of efflux pump repressors (*mexZ*), mediated by m^6^A enrichment at promoter-proximal sites [79]. Network modeling identified hub methylation nodes, such as Dam and RlmN, that act as epigenetic “master switches” of adaptation. Similarly, in *Candida albicans*, a multi-omic correlation of histone modifications and transcriptomic remodeling under fluconazole stress revealed that the histone mark H3K27ac covaries with efflux gene expression, mirroring bacterial methylome-driven activation [80]. These findings position the epigenome as a network integrator, linking metabolic cues, gene expression, and phenotypic resistance in both prokaryotic and eukaryotic pathogens.

### 6.2. Machine Learning and Predictive Epigenetic Signatures

Recent progress in AI-driven epigenomic analysis has accelerated the identification of predictive resistance signatures. Deep-learning models, such as EpiDeepRes and MethylNet, integrate methylation profiles, CRISPR perturbation datasets, and transcriptomic changes to predict antimicrobial susceptibility [81].

Trained on large datasets from third-generation sequencing platforms, these models have reported greater than 90% accuracy in distinguishing transient tolerance–like epigenetic states from stable resistance genotypes in internal cross-validation on defined isolate panels [82]. In clinical isolates of *Klebsiella pneumoniae*, machine learning classified methylome states that correlated with carbapenem minimum inhibitory concentration (MIC) values, outperforming genomic markers alone [83]. Despite these encouraging results, the robustness of current AI models across highly diverse, multidrug-resistant clinical isolates with complex mutational landscapes remains incompletely characterized. Most training and validation cohorts to date have involved relatively constrained panels of strains, and model performance can degrade when confronted with genomic backgrounds that differ substantially from those represented in the training set [82,83]. Moreover, the distinction between transient epigenetic tolerance and stable genetic resistance is typically inferred indirectly, from short-term susceptibility phenotypes and longitudinal sampling, rather than from direct experimental separation of epigenetic and genetic contributions. As a result, the real-world diagnostic accuracy of these tools for discriminating between epigenetic and genetic resistance states in routine clinical workflows remains to be rigorously evaluated. Such computational tools are transforming epigenetic research from descriptive to predictive, enabling precision diagnostics that detect resistance phenotypes before they manifest at the genetic level.

### 6.3. Integrative Multi-Omic Platforms and Databases

To facilitate cross-species comparisons, several public repositories now aggregate data on bacterial and fungal epigenomics. The EpiResistomeDB (2025) compiles over 25,000 methylome profiles from pathogens exposed to 300 antimicrobial compounds, linking methylation motifs to functional outcomes [84]. Similarly, the CRISPRepi Atlas integrates epigenome-editing perturbation data with transcriptomic and metabolomic readouts, supporting causal inference of methylation effects [35]. A comprehensive overview of key multi-omic databases and analytical platforms relevant to bacterial and fungal epigenetic studies is presented in Table 3 (see end of this section).

Multi-omic pipelines combining single-cell transcriptomics, methylomics, and proteomics are enabling cell-level resolution of resistance heterogeneity. In *Staphylococcus aureus* biofilms, single-cell methylome profiling revealed subpopulations with differential methylation at the *icaADBC* locus, which correlated with matrix production and antibiotic tolerance [79,85]. These studies underscore that cellular heterogeneity within clonal populations primarily arises from epigenetic variation, rather than genetic divergence.

### 6.4. Metabolic–Epigenetic Coupling and Feedback Regulation

Metabolism and epigenetic regulation are tightly intertwined through shared cofactors and substrates. S-adenosylmethionine (SAM), the universal methyl donor, couples central metabolism to methylation reactions [86]. Antibiotic-induced metabolic stress alters SAM flux, modulating DNA and RNA methylation rates, and thereby tuning gene expression.

In *E. coli*, depletion of SAM via inhibition of methionine adenosyltransferase (MetK) disrupted methylation of virulence and efflux genes, rendering cells hypersensitive to β-lactams [87]. Conversely, accumulation of metabolites such as fumarate and succinate can inhibit α-ketoglutarate–dependent demethylases, promoting hypermethylation and transient tolerance [68,88]. In fungi, metabolic remodeling under antifungal stress redirects acetyl-CoA toward histone acetylation, enhancing the expression of resistance genes [80]. This metabolic–epigenetic feedback constitutes a universal adaptive axis across kingdoms, one that can be pharmacologically targeted by modulating cofactors or metabolic adjuvants [89].

### 6.5. Network-Based Therapeutic Target Identification

Integrative network analysis identifies epigenetic bottlenecks, key nodes whose perturbation collapses resistance circuits. Graph-theoretical modeling of the *Pseudomonas* epigenome identified Dam, RlmN, and global regulators, such as H-NS, as central hubs connecting stress, replication, and resistance pathways [79,81]. Targeting these hubs with small molecules or CRISPR interference disrupts network stability, sensitizing bacteria to antibiotics [90]. Similarly, in *Candida* and *Aspergillus* species, epigenetic network reconstruction revealed that simultaneous inhibition of HDAC (Hda1) and histone methyltransferase (Set1) produced a synergistic effect, increasing susceptibility to azoles and echinocandins [23,80]. These insights support the network medicine concept that complex resistance phenotypes can be effectively neutralized through multi-target epigenetic modulation rather than single-gene disruption.

### 6.6. Microbiome-Wide Epigenetic Interactions

The microbial epigenome extends beyond single species, influencing entire microbial communities. In polymicrobial infections or the gut microbiota, interspecies communication occurs via epigenetically controlled metabolites and nucleic acid signals, including quorum-sensing autoinducers whose synthesis is linked to SAM- and acetyl-CoA–dependent pathways, methylated oligonucleotides released into the extracellular milieu, and membrane vesicles that package methylated DNA and small RNAs [84,89]. Epigenetic crosstalk has been observed between commensal and pathogenic bacteria, where methylated DNA fragments or vesicle-associated nucleic acids released from resistant strains induce tolerance programs in neighboring susceptible cells and promote biofilm stabilization [90]. Multi-omics mapping of the human gut microbiome reveals that antibiotic exposure triggers widespread methylome remodeling in both commensals and opportunistic pathogens, reshaping community structure and resilience [85]. Proof-of-concept community models further suggest that pharmacologic disruption of these epigenetically governed interactions, for example, quorum-sensing inhibitors or nucleases that degrade extracellular methylated DNA, can dampen biofilm formation and reduce the emergence or spread of resistant subpopulations. However, such strategies remain to be validated in vivo [90]. These findings highlight the ecological dimension of epigenetic resistance, underscoring that antibiotic stewardship must account for community-level adaptation rather than individual species responses.

### 6.7. Integrative Systems Biology: From Models to Medicine

The fusion of systems biology and epigenomics provides a framework for rational antimicrobial design. By integrating methylome data, gene regulation networks, metabolic flux, and host immune dynamics, researchers can simulate drug responses and predict the emergence of resistance [77,81]. Computational modeling platforms such as EpiNetSim now simulate the propagation of methylation signals across regulatory networks under antibiotic pressure [83]. Coupling these models with clinical microbiome sequencing enables in silico testing of epigenetic adjuvant combinations before clinical trials, thereby reducing costs and the risk of failure [84]. As artificial intelligence becomes increasingly integrated into epigenomic data interpretation, precision antimicrobial therapy is shifting toward data-driven modulation of reversible regulatory states, the hallmark of a true epigenetic medicine.

## 7. Future Directions: Toward an Epigenetic Era in Antimicrobial Therapy

The emerging view of antibiotic resistance as an epigenetically regulated phenomenon redefines the traditional boundaries of microbiology and pharmacology. For decades, resistance was attributed solely to stable genetic mutations and horizontal gene transfer. However, accumulating evidence reveals that bacteria and fungi employ dynamic, reversible epigenetic mechanisms, such as methylation, acetylation, chromatin remodeling, and RNA modification, to fine-tune gene expression, regulate stress responses, and adapt to antimicrobial pressure without altering the DNA sequence [91,92]. Rather than reiterating the mechanistic foundations and current applications of CRISPR-based epigenetic editing summarized in Section 4 and Section 5, we highlight here several forward-looking directions. These include the development of closed-loop CRISPR epigenetic systems that couple real-time sensing of resistance-associated methylation states or transcriptional signatures to on-demand editing; microbiome-aware interventions that selectively reprogram pathogens while preserving beneficial commensals; and integration with AI-driven predictive models to prioritize epigenetic targets and optimize dosing schedules. Together, these approaches outline a roadmap from current proof-of-concept studies toward programmable, patient-tailored epigenetic therapies.

### 7.1. Reframing Resistance as a Reversible and Programmable Process

Unlike mutational resistance, which is permanent, epigenetic resistance represents a reversible adaptive state that can be modulated by environmental factors and therapeutics. Third-generation sequencing has exposed methylome landscapes that fluctuate in response to antibiotic stress, generating phenotypic diversity within clonal populations [93]. These transient states provide a bet-hedging strategy, enabling a fraction of the population to survive while maintaining the genetic blueprint of susceptibility. The capacity to reprogram these reversible marks through CRISPR epigenetic editing or pharmacological intervention introduces a new dimension of control, transforming pathogens from immutable adversaries into adaptable systems that can be recalibrated to become sensitive to specific targets.

### 7.2. Integrating Multi-Omics for Precision Epigenetic Medicine

Future antimicrobial research must move beyond single-gene studies toward multi-omic integration, combining methylomics, transcriptomics, metabolomics, and proteomics into unified predictive models [94]. Machine learning and systems biology approaches can decipher the interplay between metabolic flux, methylation states, and transcriptional activity, revealing actionable nodes for intervention. Integrative frameworks, such as EpiNetSim and EpiResistomeDB, already enable the computational modeling of resistance circuits and the prediction of drug responses [84,95]. Coupled with clinical metagenomics, these systems can identify methylation biomarkers of tolerance in patient isolates, guiding the personalized selection of epigenetic adjuvants. Importantly, these biomarkers are currently used primarily as correlative indicators of drug response rather than as proven causal drivers of resistance, underscoring the need for longitudinal and interventional studies that directly test how specific epigenetic modifications influence clinical outcomes.

### 7.3. Translational Horizons: Epigenetic Adjuvants and Host Modulation

Therapeutic strategies will increasingly merge pathogen-directed and host-directed epigenetic interventions. Bacterial methyltransferase inhibitors, histone deacetylase inhibitors, and RNA-modifying enzyme inhibitors have demonstrated efficacy in re-sensitizing resistant pathogens [55,59,64]. In parallel, host epigenetic modulation, such as enhancing macrophage-trained immunity via H3K4me3 enrichment or histone acetylation, fortifies innate defenses without the need for antibiotic overuse [66,67]. These synergistic approaches align with precision medicine principles, offering sustainable antimicrobial control that minimizes selective pressure and disrupts the microbiome. The ultimate therapeutic goal is not eradication, but rather functional reprogramming, a shift from lethal targeting to metabolic and transcriptional normalization. Translating these epigenetic adjuvant and host-modulating strategies into the clinic will also require robust ethical and biosafety frameworks. In addition to conventional toxicology and pharmacokinetic profiling, longitudinal monitoring of host epigenetic regulation, immune memory, and microbiome composition will be necessary to ensure that short-term gains in pathogen control are not offset by long-term perturbations of immune and barrier homeostasis. These safeguards, discussed in more detail in Section 7.5, should be integrated into the trial design and regulatory evaluation early.

### 7.4. Synthetic Biology and Epigenetic Circuit Design

Synthetic biology will accelerate the translation of epigenetic insights into engineered interventions. Programmable CRISPR–MTase and CRISPR–TET1 systems can install or erase methylation marks with base-level precision, while bioengineered nanocarriers ensure targeted delivery to infection sites [39,70,71]. Future designs may integrate biosensing modules that monitor epigenetic states in real-time, triggering feedback-controlled drug release. Such closed-loop epigenetic therapeutics could autonomously adjust dosing to match microbial state transitions, thereby reducing toxicity and the evolution of resistance [96]. Moreover, synthetic consortia incorporating engineered commensals with “anti-resistance” epigenomes could act as living biotherapeutics, thereby restoring microbiome balance and suppressing the propagation of resistance [97]. Given their programmable and potentially self-propagating nature, epigenetic circuit designs and engineered microbial consortia also raise distinctive biosafety and ethical issues. Clinical or environmental deployment should incorporate multilayered safety features, including fail-safe kill switches, auxotrophic dependencies, and physical or ecological containment strategies, to minimize unintended persistence or spread. Governance frameworks will need to explicitly consider dual-use risks, impacts on non-target microbial communities, and the possibility of epigenetic information being transferred to bystander organisms.

### 7.5. Ethical and Biosafety Considerations

The ability to rewrite microbial epigenomes also raises ethical and ecological concerns. The dissemination of CRISPR-based epigenetic circuits in natural ecosystems could alter microbial community dynamics and rates of horizontal gene transfer [50,72]. Regulatory frameworks must evolve to ensure containment, reversibility, and environmental safety of epigenetic interventions [98]. An international consensus on bioethical governance is needed before deploying these technologies in open systems or clinical microbiomes.

### 7.6. The Epigenetic Future of Antimicrobial Discovery

Epigenetics is poised to become the cornerstone of next-generation antimicrobial discovery. Rather than identifying compounds that kill microbes, the focus will shift to molecules that reconfigure their regulatory logic, deactivate virulence, restore susceptibility, and stabilize host–microbiome symbiosis. The integration of third-generation sequencing, CRISPR epigenome editing, and artificial intelligence will enable the development of precision antimicrobial therapeutics tailored to the unique epigenetic fingerprints of each infection [99]. The translational continuum from fundamental epigenetic discoveries to clinical and biotechnological applications is illustrated in Figure 4, summarizing the stepwise progression from biomarker identification to precision epigenetic intervention. Thus, while an epigenetic perspective is reshaping the conceptual landscape of antimicrobial discovery, the concrete therapeutic interventions outlined here should be viewed as medium- to long-term goals that hinge on careful stepwise progression from proof-of-concept in vitro studies to rigorous preclinical and clinical evaluation.

By acknowledging resistance as a dynamic and reversible phenomenon, the field enters an era where the ultimate question is no longer how to kill pathogens, but how to retrain them toward coexistence. This paradigm shift represents the most profound transformation in microbiology since the discovery of antibiotics, ushering in the era of epigenetic antimicrobial therapy [100].

## 8. Conclusions

Antibiotic resistance, once viewed primarily as a consequence of genetic mutations, is now recognized as a far more complex phenomenon shaped by epigenetic adaptability. This review highlights how methylation, acetylation, and RNA modifications orchestrate rapid microbial adaptation through non-genetic reprogramming of gene expression. Advances in third-generation sequencing and methylome profiling have uncovered dynamic regulatory motifs, phase-variable methylation patterns, Dam- and Dcm-mediated networks, and histone-like acetylation systems that collectively fine-tune bacterial and fungal tolerance mechanisms. These findings underscore that the epigenome acts not merely as a passive layer of regulation but as an active modulator of resistance phenotypes, complementing rather than replacing the foundational role of genetic determinants, especially in clinical settings where epigenetic contributions are still being causally defined.

Emerging CRISPR-based epigenetic editing tools represent a transformative step toward precision antimicrobial strategies. By enabling reversible, locus-specific modification of resistance determinants, the dCas9–TET1 and dCas9–Dam systems offer a new class of programmable therapies capable of restoring drug susceptibility without compromising genomic stability. When integrated with multi-omic and AI-driven modeling, these approaches can decode regulatory hierarchies across species and predict responsiveness to epigenetic reprogramming, paving the way for individualized anti-resistance interventions. Equally important, any future clinical deployment of epigenetic adjuvants will require rigorous evaluation of host epigenome perturbation and microbiome dysbiosis, to ensure that pathogen-directed benefits are not offset by long-term collateral effects on immune and barrier homeostasis.

Looking forward, the convergence of epigenomics, systems biology, and synthetic biology will define the next generation of antimicrobial research. Translational applications will increasingly rely on CRISPR-guided epitherapeutics, epidrug combinations, and host-immunity training to achieve durable microbial control. Beyond combating resistance, the same frameworks hold promise for modulating beneficial microbiota and optimizing vaccine responses. Ultimately, bridging basic methylome science with applied therapeutic design marks a paradigm shift, from treating pathogens as static genomes to viewing them as dynamically regulated systems that can be predictively reprogrammed.

## Figures and Tables

**Figure 1 pathogens-14-01267-f001:**
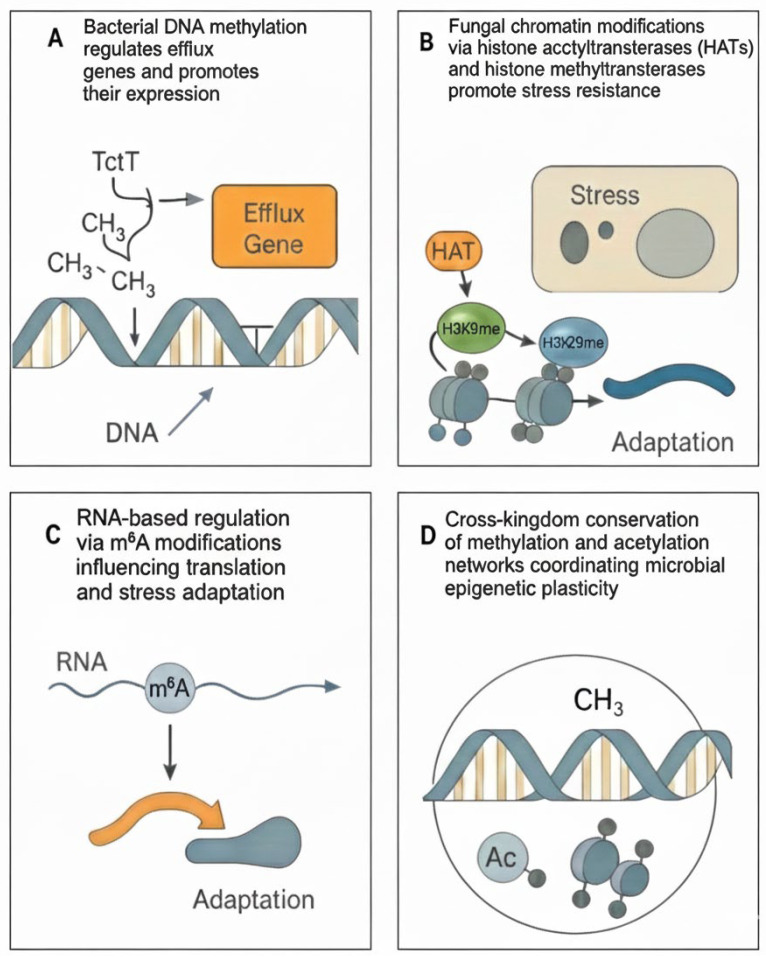
Epigenetic mechanisms underlying antibiotic resistance across bacteria and fungi. (**A**) Bacterial DNA methylation regulates the expression of efflux genes. (**B**) Fungal chromatin modifications via histone acetyltransferases (HATs) and histone methyltransferases promote stress resistance. (**C**) RNA-based regulation via m^6^A modifications influencing translation and stress adaptation. (**D**) Cross-kingdom conservation of methylation and acetylation networks coordinating microbial epigenetic resilience.

**Figure 2 pathogens-14-01267-f002:**
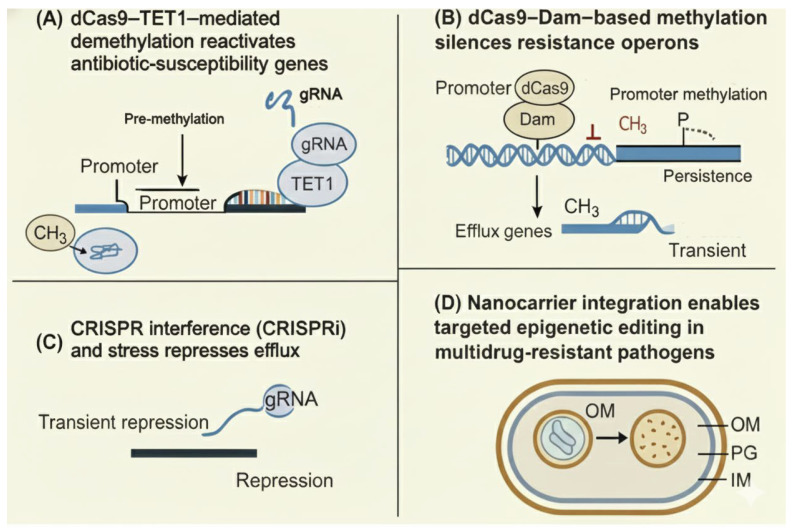
CRISPR-based epigenetic editing strategies to modulate antibiotic susceptibility. (**A**) dCas9–TET1–mediated demethylation reactivates antibiotic-susceptibility genes. A guide RNA (gRNA) directs the catalytically inactive Cas9 (dCas9) fused to the TET1 demethylase to methylated promoter regions, where removal of methyl groups (CH_3_) restores transcription of susceptibility genes. (**B**) dCas9–Dam–based methylation silences resistance operons. A gRNA directs dCas9 fused to the Dam DNA methyltransferase to resistance gene promoters, leading to promoter methylation, repression of efflux operons, and transient or persistent reduction in resistance. (**C**) CRISPR interference (CRISPRi) and stress represses efflux gene expression. A dCas9–gRNA complex sterically blocks transcription at target promoters, causing transient repression of efflux pumps and increased antibiotic sensitivity. (**D**) Nanocarrier integration enables targeted delivery of epigenetic editors to multidrug-resistant bacteria, facilitating localized CRISPR-based epigenetic modulation within the bacterial cell envelope. Abbreviations: CH_3_, methyl group; CRISPR, clustered regularly interspaced short palindromic repeats; CRISPRi, CRISPR interference; dCas9, catalytically dead Cas9; Dam, DNA adenine methyltransferase; gRNA, guide RNA; IM, inner membrane; OM, outer membrane; PG, peptidoglycan layer; TET1, ten–eleven translocation 1 demethylase.

**Figure 3 pathogens-14-01267-f003:**
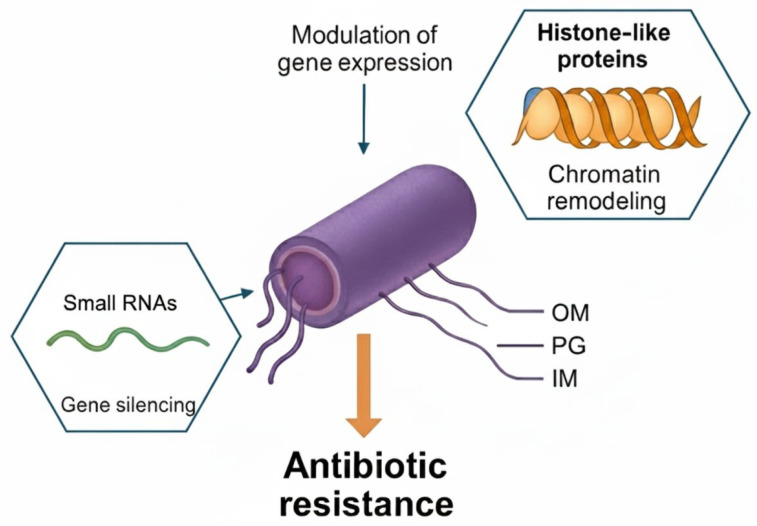
Bacterial epigenetic regulation of antibiotic resistance by small RNAs and histone-like proteins. Small non-coding RNAs (small RNAs) contribute to gene silencing by base-pairing with target transcripts and modulating mRNA stability or translation. Histone-like proteins remodel chromatin-like nucleoid structure and alter accessibility of resistance-associated genes. Together, these RNA- and protein-based epigenetic regulators modulate gene expression programs that promote antibiotic resistance in Gram-negative bacteria. Abbreviations: IM, inner membrane; OM, outer membrane; PG, peptidoglycan layer; sRNA, small RNA.

**Figure 4 pathogens-14-01267-f004:**
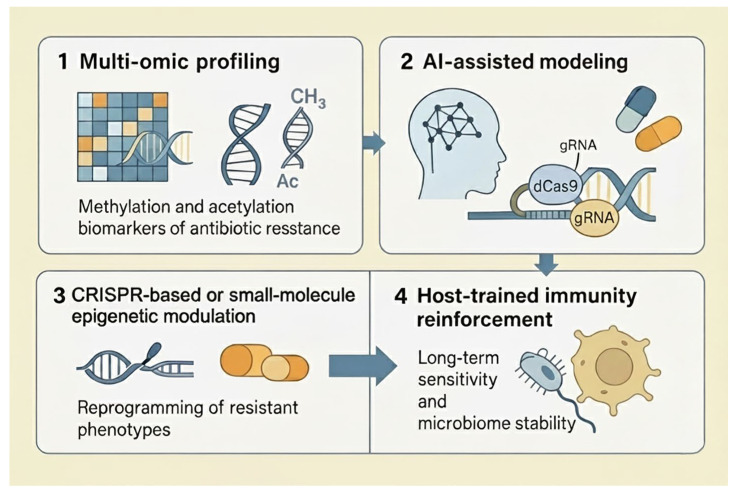
Conceptual framework for precision epigenetic therapies against antibiotic resistance. (**1**) Multi-omic profiling integrates genomics, transcriptomics, and epigenomics to identify methylation and acetylation biomarkers associated with antibiotic resistance. (**2**) AI-assisted modeling uses machine-learning algorithms to analyze these biomarkers and predict resistance phenotypes and optimal intervention points, including CRISPR targets and drug combinations. (**3**) CRISPR-based or small-molecule epigenetic modulation reprograms resistant phenotypes by editing or pharmacologically altering DNA and chromatin marks in pathogens. (**4**) Host-trained immunity reinforcement supports long-term antibiotic sensitivity and microbiome stability by leveraging epigenetic reprogramming of innate immune cells and host–microbiome interactions. Abbreviations: Ac, acetyl group; AI, artificial intelligence; CH_3_, methyl group; CRISPR, clustered regularly interspaced short palindromic repeats; dCas9, catalytically dead Cas9; gRNA, guide RNA.

**Table 1 pathogens-14-01267-t001:** Cross-kingdom epigenetic mechanisms implicated in antimicrobial resistance.

Epigenetic Mechanism	Key Molecular Players	Functional Role	Therapeutic Target/Strategy	Representative Pathogens/Systems
DNA methylation and methylome remodeling in bacteria	Dam/Dcm DNA methyltransferases; orphan MTases	Regulates virulence, efflux, biofilm formation and persistence under antibiotic pressure by altering promoter methylation and global methylome patterns	Inhibition or reprogramming of bacterial MTases; exploiting methylation signatures as biomarkers of resistance	*Escherichia coli*, *Staphylococcus aureus*, *Acinetobacter baumannii*
Phase-variable methyltransferases (“phasevarions”)	Type III and other phase-variable MTases	Generate reversible ON/OFF expression states in multiple genes (phasevarions) controlling immune evasion, niche adaptation and antibiotic tolerance	Target phase-variable MTases to freeze or redirect phasevarion states; use phasevarion signatures to stratify infections	Non-typeable *Haemophilus influenzae* and other mucosal pathogens
Chromatin-based regulation in fungi	Histone acetyltransferases (HATs), histone deacetylases (HDACs), histone methyltransferases, chromatin remodelers	Control expression of drug transporters, stress-response genes and virulence factors; drive antifungal drug resistance and tolerance	Small-molecule HAT/HDAC/HMT modulators; selective targeting of fungal chromatin enzymes (e.g., DNMT5, Hda1)	*Candida albicans*, *Candida auris*, *Cryptococcus neoformans*, other pathogenic fungi
RNAi-dependent epimutations	RNAi machinery (Dicer, Argonaute, RNA-dependent RNA polymerase); small interfering RNAs	Reversible, heritable silencing of drug-target genes and transporters, producing transient antifungal drug resistance without stable DNA mutation	Target RNAi pathways or epimutant-specific small RNAs to prevent or reverse epimutational resistance	Mucorales (e.g., *Mucor circinelloides*) and other filamentous fungi
RNA methylation and rRNA modification	rRNA methyltransferases (RlmN, Erm and related MTases); other RNA-modifying enzymes	Modify rRNA or tRNA to alter antibiotic binding or translation, contributing to resistance to linezolid, macrolides and related drugs	Develop selective inhibitors against resistance-associated RNA MTases; exploit allosteric sites to modulate their activity	*Staphylococcus aureus* and other Gram-positive and Gram-negative bacteria
Regulatory small RNAs (sRNAs) and RNA-based control	Small non-coding RNAs; RNA-binding proteins	Fine-tune transcript levels of efflux pumps, porins and stress pathways, shaping multidrug resistance and persistence	Antisense oligonucleotides or sRNA mimics/antagonists targeting resistance-associated regulatory RNAs	Multiple bacterial species including Enterobacteriaceae and *Pseudomonas* spp.
Host epigenetic reprogramming and trained immunity	Histone-modifying enzymes, DNA methyltransferases, metabolic–epigenetic circuits in innate immune cells	Induce trained immunity and altered responsiveness to secondary infections, influencing clearance of bacteria and fungi	Host-directed epigenetic therapies (e.g., HDAC inhibitors, metabolic modulators) to enhance pathogen killing while limiting immunopathology	Monocytes/macrophages and other innate immune cells in bacterial and fungal infections

**Table 2 pathogens-14-01267-t002:** Epigenetic drugs, molecular tools, and delivery platforms relevant to antimicrobial resistance.

Class/Modality	Example Compound/Tool	Primary Epigenetic Target	Reported Effect on Resistance or Infection Outcome	Model/Context
DNA methyltransferase inhibitors (cytidine analogues)	5-azacytidine, decitabine	Bacterial and fungal DNA methyltransferases (MTases); mammalian DNMTs at higher systemic doses	Reduce DNA methylation, alter expression of resistance determinants and virulence factors; can resensitize pathogens to β-lactams and aminoglycosides in experimental models	Bacterial and fungal systems; oncology-derived compounds repurposed as adjuvants in infection models
Histone deacetylase (HDAC) inhibitors	Trichostatin A, vorinostat, other pan-HDAC inhibitors	Fungal and host histone deacetylases (HDACs)	Modify chromatin accessibility and transcription; reverse or reduce antifungal resistance; in host macrophages, enhance mitochondrial ROS and bacterial clearance	*Candida* spp., *Candida auris*; human macrophages and in vitro infection models
Chromatin and histone methylation modulators	DNMT5-targeted modulators; HMT/H3K27/H3K9-directed compounds (experimental)	Fungal DNMT5 and histone methyltransferases	Maintain or disrupt high-fidelity chromatin states that underpin drug tolerance and virulence; potential to overcome stable antifungal resistance	Emerging human fungal pathogens, including *Candida auris*
CRISPR/dCas9–TET1 epigenetic editors	dCas9–TET1 fusion; targeted DNA demethylation constructs	Site-specific DNA demethylation at promoters or regulatory elements	Reactivate silenced genes (including antibiotic-susceptibility genes); enable locus-specific probing of methylation roles in resistance	Mammalian cells and microbial models using CRISPR/dCas9–TET1 editing; frameworks adaptable to pathogens
CRISPR-based antimicrobials targeting resistance genes	Cas9, Cas12, Cas13, Cas14 constructs; phage-delivered CRISPR cassettes	Resistance genes, plasmids, essential loci in pathogens	Sequence-specific cleavage or silencing of resistance determinants; selective killing or resensitization of multidrug-resistant bacteria and fungi	*E. coli*, *Acinetobacter baumannii*, *Salmonella enterica*, *Candida* spp. and other pathogens
RNA-targeting CRISPR tools	CRISPR–Cas13 systems	RNA transcripts (including resistance gene mRNAs, regulatory RNAs)	Degrade target RNAs in a sequence-specific manner; enable programmable inhibition of resistance-associated transcripts and virulence factors	Bacterial models using Cas13a-based antimicrobials and RNA-targeting constructs
Regulatory RNA-based approaches	Antisense oligonucleotides, sRNA mimics/antagonists	Resistance-associated mRNAs and sRNAs	Modulate expression of efflux pumps, porins and other resistance determinants; potential new class of RNA therapeutics against MDR bacteria	Multiple Gram-positive and Gram-negative bacteria
Nanoparticle and carrier-based epigenetic delivery	CRISPR–nanoparticle formulations; chitosan-coated nanoparticles; polymeric nanoformulations	Vehicle for CRISPR constructs, epidrugs or antimicrobial peptides	Enhance delivery to infection sites and biofilms; improve stability and local concentration of epigenetic modulators and antibiotics; reduce systemic toxicity	Biofilm-driven infections; uropathogenic *E. coli*; multidrug-resistant bacteria
AI- and multi-omics-guided epigenetic therapy design	AI-driven drug repositioning pipelines; Deep-Chrome-like models	Integration of histone marks, methylation, expression and drug-response data	Identify epigenetic biomarkers and candidate epidrugs; prioritize CRISPR/epigenetic targets and reposition existing drugs	Oncology and epigenetic therapy models; conceptual framework extendable to infectious diseases

**Table 3 pathogens-14-01267-t003:** Representative examples of epigenetic regulation and epigenetic-based therapeutic strategies in antimicrobial resistance.

Epigenetic Mechanism	Microbial System/Context	Resistance or Therapeutic Phenotype	Key References
DNA methylation and methylome remodeling	*Staphylococcus aureus* hospital isolates; *Acinetobacter baumannii* clinical strains	Altered genomic methylation promotes environmental persistence and antibiotic tolerance in *S. aureus*; combined methylome–transcriptome analysis in *A. baumannii* reveals a methylation-dependent, epigenetic-based mechanism of antibiotic resistance	[11,54]
Phase-variable methyltransferases (“phasevarions”)	Non-typeable *Haemophilus influenzae*	A biphasic epigenetic switch driven by phase-variable methyltransferase controls immunoevasion, virulence and niche adaptation, indirectly shaping antibiotic tolerance	[7,18,77]
Fungal chromatin remodeling	*Candida* spp. and *Candida auris*	Histone acetylation and methylation pathways regulate expression of efflux pumps and stress genes; targeting epigenetic regulators overcomes azole and echinocandin resistance in *C. auris*	[5,6,19,24,61,81]
RNAi-dependent epimutations	Mucorales and other filamentous fungi	RNAi-dependent epimutations transiently and heritably silence drug-target genes, conferring reversible antifungal resistance that disappears when drug pressure is removed	[17,21,25]
RNA methyltransferases and rRNA modification	*Staphylococcus aureus* RlmN and Erm MTases	Inactivation of RlmN increases linezolid resistance; allosteric regulation of Erm rRNA methyltransferases modulates macrolide–lincosamide resistance, identifying enzyme surfaces as potential drug targets	[64,65]
Regulatory small RNAs	Diverse bacterial pathogens	sRNAs and other regulatory RNAs modulate expression of efflux pumps, porins and stress pathways, contributing to multidrug resistance and persistence; proposed as novel drug targets	[13,45,70]
CRISPR-based antimicrobials targeting resistance determinants	*E. coli*, *Acinetobacter baumannii*, *Salmonella enterica* and other bacteria	CRISPR–Cas-based antimicrobials (including Cas13a) selectively cleave resistance genes or essential loci, enabling sequence-specific killing or resensitization of multidrug-resistant bacteria	[15,35,37,44,47,76,84]
CRISPR-based genome/epigenome manipulation in fungi	*Candida* species	CRISPR/Cas and CRISPR ribonucleoprotein platforms enable targeted gene and regulatory-region manipulation, providing tools to dissect and potentially reverse chromatin-mediated antifungal resistance	[39,46]
Host epigenetic reprogramming and trained immunity	Monocytes/macrophages and other innate immune cells	Epigenetic remodeling underlies trained immunity and altered responsiveness to infections; HDAC inhibitors promote mitochondrial ROS and bacterial clearance, suggesting host-directed epigenetic therapies	[28,31,60,67,68,69]
Multi-omics and AI-guided epigenetic biomarker discovery	Epigenetic biomarker and drug-repositioning studies	Third-generation sequencing and AI models enable genome-wide methylation profiling and prediction of gene expression from histone marks; multi-omics and AI-driven pipelines identify epigenetic drug targets and repositioned epidrugs as candidates for future anti-infective strategies	[3,32,34,82,83]

## Data Availability

No new data were created or analyzed in this review; all information is available in the cited publications and public repositories referenced herein. Further inquiries can be directed to the corresponding author.

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
