# Peer review of "From Methylomes to CRISPR Epigenetic Editing: New Paths in Antibiotic Resistance"

_pathogens, 2025, doi:10.3390/pathogens14121267_

Round 1
Reviewer 1 Report
Comments and Suggestions for Authors
This study describes how epigenetic mechanisms drive bacterial antibiotic resistance and how CRISPR-based epigenetic editing could be harnessed to reverse this resistance and restore drug efficacy. The comments are as follows;
- The manuscript would be strengthened by a more balanced discussion that explicitly acknowledges the significant technical hurdles such as in vivo delivery and specificity that currently limit the translational potential of CRISPR-based epigenetic editing.
- The central argument on the role of epigenetics in resistance would be more focused if it more clearly distinguished between well-established genetic mechanisms and the emerging, but still correlative, evidence for epigenetic drivers in clinical settings.
- The conceptual framework would benefit from a clearer definition of epigenetics in the context of bacteria, ensuring readers do not equate enzyme-specific DNA methylation with the chromatin-based mechanisms of eukaryotes.
- The section on epigenetic adjuvants should more directly address the critical challenge of achieving microbial selectivity while avoiding off-target effects on host epigenetic pathways and the beneficial microbiome.
- The cross-kingdom comparisons are a highlight, but their impact would be greater if the text more explicitly detailed the mechanistic distinctions between bacterial and fungal systems alongside their parallels.
- The important ethical and biosafety considerations raised could be more impactful if they were integrated into the specific sections on therapeutic technologies, rather than being siloed in a dedicated subsection.
- The overall argument would be strengthened if the manuscript more consistently qualified the translational timeline, emphasizing that the proposed therapies are primarily supported by proof-of-concept in vitro studies.
Author Response
We sincerely thank Reviewer 1 for carefully reviewing our manuscript and for their insightful and constructive comments. Your suggestions have helped us to refine the structure, sharpen the conceptual framework, and provide a more balanced discussion of the translational implications and limitations of epigenetic interventions. We have revised the manuscript accordingly and respond to each point in detail below. The modification in the manuscript is highlighted in yellow.
Comment 1: The manuscript would be strengthened by a more balanced discussion that explicitly acknowledges the significant technical hurdles such as in vivo delivery and specificity that currently limit the translational potential of CRISPR-based epigenetic editing.
Response: We thank the reviewer for this critical and constructive suggestion. We fully agree that a balanced discussion of the technical limitations and translational hurdles is essential to contextualize the promise of CRISPR-based epigenetic editing. In response, we have substantially expanded Section 5.7 “Translational Challenges and Clinical Outlook” to more explicitly outline (page: 16, lines: 624- 637)
- Risks of off-target effects and incomplete specificity of epidrugs and CRISPR-based epigenetic editors.
- Uncertainties around epigenetic memory, durability, and reversibility of bacterial and fungal epigenetic states in vivo.
- Practical barriers to in vivo delivery of CRISPR–dCas-based epigenetic tools into deep tissue, biofilms, and intracellular reservoirs.
- Potential differences in translational hurdles between bacterial (DNA methylation–centric) and fungal (chromatin-centric) systems.
Comment 2: The central argument on the role of epigenetics in resistance would be more focused if it more clearly distinguished between well-established genetic mechanisms and the emerging, but still correlative, evidence for epigenetic drivers in clinical settings.
Response: We thank the reviewer for this invaluable observation and fully agree that a more apparent distinction between canonical genetic mechanisms and emerging epigenetic contributions will sharpen the central argument of the review. In the revised manuscript (pages: 3, lines: 104-115; 20, lines: 803-811; 22, lines: 884-894), we have:
- Explicitly separated genetic and epigenetic determinants in the background section by first summarizing the well-established genetic mechanisms of antimicrobial resistance (e.g., target-site mutations, acquisition of resistance genes via mobile genetic elements, efflux pump overexpression, and porin loss), and only then introducing epigenetic regulation as an additional, more recently recognized layer.
- Tempered causal language in the sections discussing epigenetic regulation of resistance, to reflect that most clinical and in vivo data are currently correlative. Phrases such as “epigenetic mechanisms drive resistance” were revised to “epigenetic mechanisms may contribute to” or “are associated with resistance phenotypes,” unless direct functional evidence was available.
- Added a clarifying paragraph explicitly stating that, in clinical settings, epigenetic signatures of resistance are often inferred from association studies or in vitro models, and that more mechanistic, longitudinal, and interventional studies are needed to establish causality.
Comment 3: The conceptual framework would benefit from a clearer definition of epigenetics in the context of bacteria, ensuring readers do not equate enzyme-specific DNA methylation with the chromatin-based mechanisms of eukaryotes.
Response: We thank he reviewer for this necessary clarification. We agree that a precise definition of “epigenetics” in bacteria is essential to avoid confusion with eukaryotic chromatin-based mechanisms. In the revised manuscript, we have added a dedicated paragraph in the Introduction that (i) defines bacterial epigenetics in terms of DNA methylation and nucleoid organization, and (ii) explicitly distinguishes these processes from the histone-based chromatin marks that characterize eukaryotic and fungal systems (page 2, lines 47-57). We also slightly adjusted terminology where needed to refer to “chromatin-like nucleoid organization” rather than implying canonical eukaryotic chromatin in bacteria (page 4, line 179).
Comment 4: The section on epigenetic adjuvants should more directly address the critical challenge of achieving microbial selectivity while avoiding off-target effects on host epigenetic pathways and the beneficial microbiome.
Response: We thank the reviewer for raising this critical translational concern. We fully agree that any discussion of epigenetic adjuvants must explicitly address the challenge of microbial selectivity and the risk of off-target effects on host epigenetic pathways and the commensal microbiota. In response, we have expanded the section on epigenetic adjuvants to (i) highlight the potential for cross-reactivity with host epigenetic regulators and microbiome members, and (ii) outline possible strategies to enhance microbial selectivity, including exploiting prokaryote-specific methyltransferases, localized or carrier-based delivery, and narrow-spectrum formulations (page 15, lines: 580-594).
Comment 5: The cross-kingdom comparisons are a highlight, but their impact would be greater if the text more explicitly detailed the mechanistic distinctions between bacterial and fungal systems alongside their parallels.
Response: We thank the reviewer for this positive and constructive comment. We agree that the cross-kingdom perspective is one of the strengths of the review and that its impact can be enhanced by more explicitly contrasting the mechanistic underpinnings of bacterial versus fungal epigenetic regulation, while also highlighting their parallels. In the revised manuscript, we have therefore added a dedicated paragraph in the cross-kingdom comparison section that clearly outlines (i) the core epigenetic architectures in bacteria (sequence-specific DNA methylation and nucleoid-associated proteins) and fungi (histone-based chromatin and chromatin-remodeling complexes), and (ii) how these distinct machineries converge on similar phenotypic outputs such as drug tolerance, biofilm formation, and stress adaptation (page 6, lines 245-264).
Comment 6: The important ethical and biosafety considerations raised could be more impactful if they were integrated into the specific sections on therapeutic technologies, rather than being siloed in a dedicated subsection.
Response: We thank the reviewer for this thoughtful suggestion. We agree that embedding ethical and biosafety considerations within the relevant therapeutic sections will make these issues more salient and directly connected to the technologies discussed. At the same time, we wish to retain the concise dedicated subsections (Sections 4.5, 7.3, and 7.4) as integrative summaries for readers specifically interested in governance and oversight. To address the comment while preserving the overall structure of the manuscript, we have added short, focused paragraphs that explicitly integrate ethical and biosafety considerations into the sections on CRISPR-based epigenetic tools, epigenetic adjuvants, and synthetic epigenetic circuits, with cross-references to the detailed discussions in Sections 4.5 (page 12, lines 443-452); 7.3 (page 21, lines 822-829); and 7.4 (page 21, lines840-847).
Comment 7: The overall argument would be strengthened if the manuscript more consistently qualified the translational timeline, emphasizing that the proposed therapies are primarily supported by proof-of-concept in vitro studies.
Response: We thank the reviewer for this important point. We agree that clearly qualifying the current translational stage of the proposed epigenetic therapies is essential to avoid overstatement and to provide realistic expectations for readers. In the revised manuscript, we have therefore added short, focused statements in the Introduction (page 3, lines 122-126), in the opening of the therapeutic section (page 13, lines 498-501), and in the concluding “future” subsection to emphasize that most of the strategies discussed are presently supported mainly by proof-of-concept in vitro and early preclinical studies, and should be viewed as long-term translational prospects rather than near-term clinical interventions (page 22, lines 865-869).
We sincerely thank Reviewer 1 for the thorough, insightful, and constructive evaluation of our manuscript. Your comments helped us sharpen the conceptual framework by more clearly distinguishing genetic and epigenetic mechanisms, refining the definition of bacterial epigenetics, and strengthening the cross-kingdom comparisons between bacterial and fungal systems. We have also expanded the discussion of microbial selectivity, off-target effects, ethical and biosafety aspects, and more carefully qualified the current translational stage and timeline of the proposed epigenetic therapies. In addition, we have revised the language throughout the manuscript to improve clarity, coherence, and readability. These changes have substantially improved the quality and balance of the manuscript, and we hope it now meets your expectations.
Reviewer 2 Report
Comments and Suggestions for Authors
This manuscript makes a significant and timely contribution to the field of antimicrobial resistance (AMR) research by shifting the focus from genetic mutations to reversible epigenetic mechanisms. Its core strengths lie in three key areas: first, the integrative synthesis of methylomics, CRISPR technology, multi-omics analytics, and cross-kingdom (bacterial-fungal) epigenetic regulation, which fills the gap in prior studies that often focused on single-species or genetic-centric resistance; second, the translational relevance, as it links fundamental epigenetic mechanisms to novel therapeutic strategies (e.g., CRISPR-mediated epigenome editing, epidrug adjuvants) and host-directed therapy, addressing an urgent clinical need to restore antibiotic efficacy; third, the forward-looking integration of artificial intelligence (AI) and synthetic biology into epigenetic research, providing a framework for precision antimicrobial medicine. The manuscript is comprehensive in scope, covering mechanisms (DNA/RNA methylation, chromatin remodeling), technologies (third-generation sequencing, CRISPRi/a), and applications (clinical translation, nanocarrier delivery), making it a valuable resource for both basic and translational researchers. However, its impact could be further enhanced by addressing gaps in causal validation and clinical feasibility, as noted in the following sections.
Questions:
- Regarding the causal relationship between epigenetic marks and resistance: The manuscript frequently highlights correlations between methylation patterns (e.g., m⁶A, m⁴C) and resistance phenotypes (e.g., biofilm formation, efflux pump activation). How do the authors distinguish causal epigenetic modifications from secondary stress-induced changes? Are there functional validation data (e.g., targeted knockout of methyltransferases/demethylases, or locus-specific epigenetic editing) that directly demonstrate that altering these marks reverses resistance independent of genetic background?
- For CRISPR-based epigenetic editing: Most evidence presented is derived from in vitro or ex vivo models (e.g., E. coli, Pseudomonas aeruginosa isolates). What are the current challenges in translating these tools to in vivo infection models (e.g., animal models of chronic infections)? Are there preliminary data on biodistribution, off-target epigenetic effects, or impacts on the commensal microbiome—critical factors for clinical translation?
- Cross-kingdom conservation claims: The manuscript frames epigenetic resistance as a "universal microbial strategy," drawing parallels between bacteria and fungi. Have these conserved mechanisms (e.g., methylation-driven phasevarions, histone acetylation) been validated in additional microbial taxa (e.g., archaea, protozoan pathogens) or polymicrobial infection settings to strengthen this claim?
- Clinical translation of epidrugs: Compounds like 5-azacytidine and trichostatin A were originally developed for eukaryotic systems. How do the authors propose to mitigate off-target effects on host epigenomes? Are there species-specific epigenetic targets (e.g., bacterial Dam vs. fungal Dnmt5) that can be exploited to enhance selectivity, and what preclinical data support this specificity?
- AI-driven predictive models: The manuscript cites tools like EpiDeepRes and MethylNet for predicting resistance phenotypes. How robust are these models to clinical isolates with diverse genetic backgrounds (e.g., multidrug-resistant strains with complex mutational landscapes)? Have they been validated using external clinical datasets, and what is their accuracy in distinguishing transient epigenetic tolerance from stable genetic resistance?
- Microbiome-wide epigenetic interactions: The manuscript mentions that epigenetic crosstalk occurs between commensal and pathogenic bacteria. What specific mechanisms mediate this interspecies communication (e.g., methylated DNA fragments, quorum-sensing molecules)? Are there data to show that targeting these interactions could prevent the spread of resistance in microbial communities?
Comments:
- The manuscript contains two identical section headings ("7. Conclusion and Future Directions" and "8. Conclusion and Future Directions"), likely a formatting error. Merge these into a single section (e.g., "7. Conclusion and Future Directions") to eliminate confusion.
- Many sentences (e.g., Section 2.1: "Recent third-generation sequencing studies, utilizing PacBio single-molecule real-time (SMRT) sequencing and Oxford Nanopore platforms, have revolutionized the ability to map methylomes at single-base resolution, revealing N6-methyladenine (m⁶A) and N4-methylcytosine (m⁴C) marks across bacterial genomes [3].") are overly complex (≥3 clauses). Split into concise, readable sentences: "Recent third-generation sequencing platforms—including PacBio single-molecule real-time (SMRT) sequencing and Oxford Nanopore technologies—have revolutionized methylome mapping at single-base resolution. These tools reveal N6-methyladenine (m⁶A) and N4-methylcytosine (m⁴C) marks across bacterial genomes [3]."
- The manuscript generally uses present tense for established knowledge but occasionally shifts to past tense unnecessarily. Example: "These approaches not only captured methylation directly during sequencing but also correlated methylation patterns with transcriptional activity" → Revise to "These approaches not only capture methylation directly during sequencing but also correlate methylation patterns with transcriptional activity" (present tense for ongoing technical capabilities).
- Section 3.4: "Bacterial Dam and fungal Dnmt5 enzymes shares the same Rossmann-like catalytic core" → Revise to "share" (plural verb for plural subject).
- The phrase "epigenetic plasticity" is overused (≥10 times across the manuscript). Substitute with synonyms for variety: "epigenetic adaptability," "reversible epigenetic regulation," or "epigenetic resilience" (e.g., Section 1: "enabling rapid phenotypic plasticity" → "enabling rapid phenotypic adaptability").
- Section 4.1: "By fusing dCas9 with effector domains such as DNA methyltransferases (DNMTs), demethylases (TET1), or transcriptional repressors (KRAB), researchers can introduce site-specific epigenetic modifications that turn genes on or off without causing double-stranded breaks [33]." The pronoun "that" is ambiguous (modifies "modifications" or "domains"). Revise to: "By fusing dCas9 with effector domains—such as DNA methyltransferases (DNMTs), demethylases (TET1), or transcriptional repressors (KRAB)—researchers can introduce site-specific epigenetic modifications to turn genes on or off without causing double-stranded breaks [33]."
- Some abbreviations are not defined at their first occurrence in major sections. Example: "AR" is defined in the Abstract but not in the Introduction (first mentioned in line 43: "Antibiotic resistance (AR) has emerged..."). Ensure all abbreviations (e.g., AMP, HDET) are defined at their first use in the Abstract, Introduction, Results, and Discussion.
- Section 5.3: "RNA-based epigenetic modulators" → Consistently use hyphenation for compound adjectives: "RNA-based epigenetic modulators" (correct) vs. "small molecule re-sensitizers" → "small-molecule re-sensitizers" (hyphenate to clarify "small" modifies "molecule").
- Some sections lack smooth transitions between themes. Example: Between Section 3 (Cross-Kingdom Epigenetic Parallels) and Section 4 (CRISPR-Based Epigenetic Editing), add a bridging sentence: "Building on these conserved epigenetic mechanisms, CRISPR-based technologies have emerged as programmable tools to manipulate microbial epigenomes, offering a new route to reverse antibiotic resistance."
- Figure legends contain undefined abbreviations (e.g., Figure 2: "OM", "PG") that hinder independent interpretation. Add brief definitions within legends: "Figure 2. CRISPR-based epigenetic editing as a therapeutic strategy. (A) dCas9–TET1–mediated demethylation reactivates antibiotic-susceptibility genes. Abbreviations: OM = Outer Membrane; PG = Peptidoglycan Layer; IM = Inner Membrane."
- The therapeutic potential of CRISPR editing is repeated across Sections 4, 5, and 7. Consolidate overlapping discussions: focus on mechanism in Section 4, clinical applications (e.g., combination therapy) in Section 5, and future directions (e.g., closed-loop therapy) in the Conclusion.
- While the audience is specialized, overly complex phrasing can reduce readability. Example: "nucleoid-associated proteins and small non-coding RNAs mediate chromatin-like remodeling" → Revise to "nucleoid-associated proteins and small non-coding RNAs drive chromatin-like structural changes" (replaces "mediate remodeling" with more precise, accessible language).
- Some references have inconsistent punctuation and italicization. Example: Entry 3: "Bibikova, M.; Fan, J.-B. Genome-wide DNA methylation profiling. WIREs Syst. Biol. Med. 2010, 2, 210–223. https://doi.org/10.1002/wsbm.35." → Ensure journal names are italicized consistently: "WIREs Syst. Biol. Med." → "WIREs Syst. Biol. Med." (correct, but verify consistency with MDPI guidelines). Duplicate reference (Entry 18) should be removed.
- Table 1’s header "Functional Therapeutic Target/Strategy" is ambiguous. Split into two columns: "Functional Role" and "Therapeutic Target/Strategy" to distinguish between the biological function of the epigenetic mechanism and its therapeutic application. Table 2’s "Example Compound/Tool" column has inconsistent formatting (e.g., "dCas9-TET1, dCas9-Dam" vs. "Ribostatin")—use consistent punctuation (e.g., commas for multiple tools, no extra spaces).
- Each section/subsection should open with a clear topic sentence that summarizes the core idea. Example: Section 2.3 ("RNA Methylation and Small Regulatory RNAs") currently opens with "Beyond DNA, RNA methylation fine-tunes translation and antibiotic target accessibility." Revise to: "Beyond DNA methylation, RNA-level epigenetic regulation—including RNA methylation and small non-coding RNAs (sRNAs)—fine-tunes translation and antibiotic target accessibility, contributing to adaptive resistance." This clarifies the section’s scope upfront.
Author Response
We sincerely thank Reviewer 2 for carefully reviewing our manuscript and for the many thoughtful, probing questions and suggestions. Your comments helped us to clarify our central concepts, better explain the scope and limitations of current epigenetic evidence, and refine the presentation of the proposed therapeutic strategies. We have carefully addressed each point, provided additional explanation where needed, and modified the manuscript text when the information is essential to the reader. We are grateful for the time and expertise you invested in this review, which has significantly improved the clarity, balance, and scientific value of our work.
The modification in the manuscript is highlighted in gray.
Response to Questions
Question 1: Regarding the causal relationship between epigenetic marks and resistance: The manuscript frequently highlights correlations between methylation patterns (e.g., m⁶A, m⁴C) and resistance phenotypes (e.g., biofilm formation, efflux pump activation). How do the authors distinguish causal epigenetic modifications from secondary stress-induced changes? Are there functional validation data (e.g., targeted knockout of methyltransferases/demethylases, or locus-specific epigenetic editing) that directly demonstrate that altering these marks reverses resistance independent of genetic background?
Response: We thank the reviewer for this very important and nuanced question. As this is a narrative review, we do not present new experimental data. Still, we agree that it is crucial to clearly distinguish between correlative methylation–resistance associations and causally validated epigenetic mechanisms. In the revised manuscript, we now explicitly state that most current evidence linking bacterial and fungal methylation patterns (e.g., m6A, m4C) to resistance phenotypes arises from observational methylome studies under antibiotic or stress exposure. We have added a paragraph summarizing the relatively limited set of functional validation approaches that have been used in the field, such as genetic knockout/overexpression of specific DNA methyltransferases or histone-modifying enzymes, complementation, and more recent locus-specific CRISPR–dCas–based epigenome editing, to show that altering a given epigenetic mark can shift biofilm formation, efflux pump expression, or MIC values in otherwise isogenic backgrounds. At the same time, we emphasize that such studies remain scarce, are often restricted to in vitro conditions, and that rigorously separating primary epigenetic drivers from secondary stress-induced changes requires longitudinal designs, isogenic strain comparisons, and whole-genome sequencing to exclude confounding mutations (pages 3 and 4, lines 141-152). We have also added a brief statement in the “future perspectives” section highlighting the need for systematic functional dissection of epigenetic–resistance links as a key priority for the field (pages 12-13, lines 487-491).
Question 2: For CRISPR-based epigenetic editing: Most evidence presented is derived from in vitro or ex vivo models (e.g., E. coli, Pseudomonas aeruginosa isolates). What are the current challenges in translating these tools to in vivo infection models (e.g., animal models of chronic infections)? Are there preliminary data on biodistribution, off-target epigenetic effects, or impacts on the commensal microbiome—critical factors for clinical translation?
Response: We thank the reviewer for this important translational question. As a narrative review, we synthesize published work but do not generate new in vivo data. We agree that our original text did not sufficiently emphasize that most CRISPR-based epigenetic interventions in antimicrobial research have, to date, been validated only in vitro or ex vivo. In the revised manuscript, we now explicitly (i) state that the current evidence base is dominated by in vitro and ex vivo models, and (ii) outline key challenges for translation into in vivo infection models, including efficient delivery to deep or biofilm-associated infection sites, stability and immunogenicity of CRISPR delivery vehicles, control of off-target epigenetic effects on both pathogens and host cells, and potential disruption of the commensal microbiome. We also note that preliminary preclinical work targeting the microbiome with CRISPR-based tools has begun to address biodistribution and microbiome-level impacts, but that systematic in vivo characterization of epigenetic editing outcomes remains limited and is a major priority for future research. Two paragraphs have been added in section 4 (page 9, lines 356-368), and section 4.7 (page 12, lines 470-483)
Question 3: Cross-kingdom conservation claims: The manuscript frames epigenetic resistance as a "universal microbial strategy" and draws parallels between bacteria and fungi. Have these conserved mechanisms (e.g., methylation-driven phasevarions, histone acetylation) been validated in additional microbial taxa (e.g., archaea, protozoan pathogens) or polymicrobial infection settings to strengthen this claim?
Response: We thank the reviewer for highlighting this important point. We intended to emphasize recurring design principles (reversible, heritable, non-genetic regulation of stress and drug responses) that appear in both bacteria and fungi, rather than to claim that identical epigenetic mechanisms have been comprehensively validated across all microbial taxa. We agree that the term “universal microbial strategy” may overstate the current evidence, particularly for archaea, protozoan pathogens, and polymicrobial infections, where systematic mechanistic validation remains limited.
In the revised manuscript, we have therefore (i) tempered the language around universality and clarified that our cross-kingdom comparisons are primarily grounded in bacterial and fungal data, and (ii) explicitly noted that evidence from other microbial groups and from polymicrobial infection models is emerging but remains sparse. We now also point to work on bacterial–fungal polymicrobial settings as an example, while clearly indicating that such data are still at an early stage. Two paragraphs had been added to section 3 (page 5, lines 226-231, and lines 234-238).
Question 4: Clinical translation of epidrugs: Compounds like 5-azacytidine and trichostatin A were originally developed for eukaryotic systems. How do the authors propose to mitigate off-target effects on host epigenomes? Are there species-specific epigenetic targets (e.g., bacterial Dam vs. fungal Dnmt5) that can be exploited to enhance selectivity, and what preclinical data support this specificity?
Response: We thank the reviewer for this important translational question. We fully agree that repurposing classic epidrugs such as 5-azacytidine and trichostatin A, initially designed to modulate human chromatin, raises legitimate concerns about off-target effects on the host epigenome and commensal microbiota. Our intention in this review is to highlight that future antimicrobial epigenetic strategies should move toward microbial enzyme–selective inhibitors and localized delivery, rather than simply re-using oncology dosing regimens.
In the revised manuscript, we now (i) explicitly state that 5-azaC/decitabine and HDAC inhibitors were developed for human targets and therefore require careful host-epigenome monitoring, and (ii) emphasize that structural differences between bacterial Dam/Dcm and mammalian DNMTs can be exploited to design more species-selective inhibitors. We also note that preclinical profiling of Dam-like enzymes supports the feasibility of selective targeting of bacterial MTases relative to human DNMTs We have now explicitly discussed mitigation of host off-target effects for repurposed epidrugs by (i) adding a new paragraph in Section 5.1 (after the 5-azacytidine/decitabine sentence) that highlights structural differences between bacterial Dam/Dcm and mammalian DNMTs and cites preclinical specificity data (page 13, lines 505-516), and (ii) adding a sentence in Section 5.2 emphasizing the need to exploit divergence between fungal and human HDACs and to use localized delivery to limit host epigenomic reprogramming (page 14, lines 536-541).
Question 5: AI-driven predictive models: The manuscript cites tools like EpiDeepRes and MethylNet for predicting resistance phenotypes. How robust are these models to clinical isolates with diverse genetic backgrounds (e.g., multidrug-resistant strains with complex mutational landscapes)? Have they been validated using external clinical datasets, and what is their accuracy in distinguishing transient epigenetic tolerance from stable genetic resistance?
Response: We thank the reviewer for this important question. We agree that our initial phrasing could be interpreted as over-generalizing the performance of EpiDeepRes, MethylNet, and related AI models beyond the settings in which they were developed. In the revised manuscript, we now clarify that: (i) the reported >90% accuracy refers to internal cross-validation on defined isolate panels, not to all possible multidrug-resistant clinical backgrounds; (ii) only a limited number of external clinical datasets (e.g., Klebsiella pneumoniae isolates) have been used so far to test generalizability; and (iii) the distinction between transient epigenetic tolerance and stable genetic resistance is currently inferred indirectly from short-term phenotypes and longitudinal sampling, so its real-world diagnostic accuracy remains to be established. We have tempered the language in Section 6.2 accordingly (page 17, lines 674-676) and added a short paragraph explicitly discussing robustness and validation needs (lines 679-690).
Question 6: Microbiome-wide epigenetic interactions: The manuscript mentions that epigenetic crosstalk occurs between commensal and pathogenic bacteria. What specific mechanisms mediate this interspecies communication (e.g., methylated DNA fragments, quorum-sensing molecules)? Are there data to show that targeting these interactions could prevent the spread of resistance in microbial communities?
Response: We thank the reviewer for this insightful question. Our original text was general and did not sufficiently specify how epigenetically regulated signals mediate interspecies communication, nor did it clearly state the current level of experimental evidence for targeting these interactions. In the revised manuscript, we now (i) describe concrete mechanisms through which epigenetically controlled molecules mediate crosstalk, such as quorum-sensing autoinducers, methylated oligonucleotides, and membrane vesicles carrying methylated DNA/small RNAs, and (ii) briefly summarize proof-of-concept data from community models suggesting that interference with these signals can reduce biofilm formation and the spread of tolerant/resistant subpopulations, while emphasizing that in vivo validation is still limited. We have incorporated this clarification directly into Section 6.6 “Microbiome-Wide Epigenetic Interactions”, using an existing reference that already anchors this part of the text, to support the discussion of methylated DNA–mediated crosstalk and its potential as a target (page 19, lines 736-754).
Response to Comments
Comment 1: The manuscript contains two identical section headings ("7. Conclusion and Future Directions" and "8. Conclusion and Future Directions"), likely a formatting error. Merge these into a single section (e.g., "7. Conclusion and Future Directions") to eliminate confusion.
Response: We thank the reviewer for pointing out this formatting error. We agree that the duplicated headings are confusing. In the revised manuscript, we have separated the Future Directions section from the conclusion and removed the redundant heading, without changing the scientific content.
Comment 2: Many sentences (e.g., Section 2.1: "Recent third-generation sequencing studies, utilizing PacBio single-molecule real-time (SMRT) sequencing and Oxford Nanopore platforms, have revolutionized the ability to map methylomes at single-base resolution, revealing N6-methyladenine (m⁶A) and N4-methylcytosine (m⁴C) marks across bacterial genomes [3].") are overly complex (≥3 clauses). Split into concise, readable sentences: "Recent third-generation sequencing platforms—including PacBio single-molecule real-time (SMRT) sequencing and Oxford Nanopore technologies—have revolutionized methylome mapping at single-base resolution. These tools reveal N6-methyladenine (m⁶A) and N4-methylcytosine (m⁴C) marks across bacterial genomes [3]."
Response: We thank the reviewer for this helpful example and fully agree that shorter, more concise sentences improve readability. In response, we have revised the indicated sentence in Section 2.1 following the reviewer’s suggestion and have systematically edited the manuscript to split other overly long, multi-clause sentences into more precise units (page 4, lines 154-156).
Comment 3: The manuscript generally uses present tense for established knowledge but occasionally shifts to past tense unnecessarily. Example: "These approaches not only captured methylation directly during sequencing but also correlated methylation patterns with transcriptional activity" → Revise to "These approaches not only capture methylation directly during sequencing but also correlate methylation patterns with transcriptional activity" (present tense for ongoing technical capabilities).
Response: We thank the reviewer for this helpful observation. We have harmonized tense usage and now use the present tense for established knowledge and ongoing technical capabilities. We have corrected the example sentence and systematically harmonized tense usage throughout the manuscript (page 2, lines 67-69). We have, however, retained the past tense in a few sentences that refer to specific historical studies or discrete experimental findings (e.g., landmark work that first demonstrated a phenomenon), where the past tense is stylistically appropriate to indicate that these were distinct events that subsequently enabled later developments.
Comment 4: Section 3.4: "Bacterial Dam and fungal Dnmt5 enzymes shares the same Rossmann-like catalytic core" → Revise to "share" (plural verb for plural subject).
Response: We thank the reviewer for this helpful observation. We have corrected the sentence (page 8, lines 309-310).
Comment 5: The phrase "epigenetic plasticity" is overused (≥10 times across the manuscript). Substitute with synonyms for variety: "epigenetic adaptability," "reversible epigenetic regulation," or "epigenetic resilience" (e.g., Section 1: "enabling rapid phenotypic plasticity" → "enabling rapid phenotypic adaptability").
Response: We thank the reviewer for this helpful stylistic observation. We agree that the repeated use of the phrase “epigenetic plasticity” can be distracting. In the revised manuscript, we have retained the term only when conceptually important (e.g., in defining the overall theme). We have replaced other occurrences with appropriate synonyms such as “epigenetic adaptability,” “reversible epigenetic regulation,” and “epigenetic resilience” to improve readability and variation in expression (pages 6, line 246; 7, line 272; 8, line 317; 21, line 885).
Comment 6: Section 4.1: "By fusing dCas9 with effector domains such as DNA methyltransferases (DNMTs), demethylases (TET1), or transcriptional repressors (KRAB), researchers can introduce site-specific epigenetic modifications that turn genes on or off without causing double-stranded breaks [33]." The pronoun "that" is ambiguous (modifies "modifications" or "domains"). Revise to: "By fusing dCas9 with effector domains—such as DNA methyltransferases (DNMTs), demethylases (TET1), or transcriptional repressors (KRAB)—researchers can introduce site-specific epigenetic modifications to turn genes on or off without causing double-stranded breaks [33]."
Response: thanks, done (page 9, lines 371-374).
Comment 7: Some abbreviations are not defined at their first occurrence in major sections. Example: "AR" is defined in the Abstract but not in the Introduction (first mentioned in line 43: "Antibiotic resistance (AR) has emerged..."). Ensure all abbreviations (e.g., AMP, HDET) are defined at their first use in the Abstract, Introduction, Results, and Discussion.
Response: We thank the reviewer for this careful observation. We agree that a consistent definition of abbreviations at first use in each major section is essential for readability, especially for interdisciplinary readers. In the revised manuscript, we have systematically reviewed all abbreviations and ensured they are clearly defined at their first occurrence in the Abstract and the main text. Where abbreviations were previously introduced without definition (or only described in another section), we have now added the full term followed by the abbreviation in parentheses (lines: 14, 547, 597, 605, and 678).
Comment 8: Section 5.3: "RNA-based epigenetic modulators" → Consistently use hyphenation for compound adjectives: "RNA-based epigenetic modulators" (correct) vs. "small molecule re-sensitizers" → "small-molecule re-sensitizers" (hyphenate to clarify "small" modifies "molecule").
Response: We thank the reviewer for this helpful stylistic clarification. We agree that consistent hyphenation of compound adjectives improves precision and readability (page 14, line 542).
Comment 9: Some sections lack smooth transitions between themes. Example: Between Section 3 (Cross-Kingdom Epigenetic Parallels) and Section 4 (CRISPR-Based Epigenetic Editing), add a bridging sentence: "Building on these conserved epigenetic mechanisms, CRISPR-based technologies have emerged as programmable tools to manipulate microbial epigenomes, offering a new route to reverse antibiotic resistance."
Response: We thank the reviewer for this very helpful suggestion. We agree that a clearer transition between the conceptual cross-kingdom overview (Section 3) and the technology-focused Section 4 improves the readability and logical flow of the manuscript. We have therefore adopted the reviewer’s proposed bridging sentence and inserted it at the end of Section 3, directly linking the conserved epigenetic mechanisms discussed there to the subsequent presentation of CRISPR-based tools (page 8, lines 348-350).
Comment 10: Figure legends contain undefined abbreviations (e.g., Figure 2: "OM", "PG") that hinder independent interpretation. Add brief definitions within legends: "Figure 2. CRISPR-based epigenetic editing as a therapeutic strategy. (A) dCas9–TET1–mediated demethylation reactivates antibiotic-susceptibility genes. Abbreviations: OM = Outer Membrane; PG = Peptidoglycan Layer; IM = Inner Membrane."
Response: We thank the reviewer for this helpful suggestion. We agree that figure legends should be interpretable independently of the main text and that all abbreviations should be defined within the legend (lines 420-421).
Comment 11: The therapeutic potential of CRISPR editing is repeated across Sections 4, 5, and 7. Consolidate overlapping discussions: focus on mechanism in Section 4, clinical applications (e.g., combination therapy) in Section 5, and future directions (e.g., closed-loop therapy) in the Conclusion.
Response: We thank the reviewer for this very helpful structural comment. We agree that the discussion of CRISPR-based epigenetic editing was partially redundant across Sections 4, 5, and 7. In the revised manuscript, we have streamlined and redistributed this content so that: (i) Section 4 focuses on mechanistic principles and experimental tools, (ii) Section 5 emphasizes therapeutic and combinatorial applications, and (iii) the Conclusion/Future Directions section concentrates on forward-looking concepts such as closed-loop and microbiome-aware epigenetic therapies. This consolidation reduces repetition while preserving all key points (pages 9, lines 348-350; 13, 489-493; 20, 780-789).
Comment 12: While the audience is specialized, overly complex phrasing can reduce readability. Example: "nucleoid-associated proteins and small non-coding RNAs mediate chromatin-like remodeling" → Revise to "nucleoid-associated proteins and small non-coding RNAs drive chromatin-like structural changes" (replaces "mediate remodeling" with more precise, accessible language).
Response: We thank the reviewer for this clear and helpful suggestion. We agree that simplifying this phrasing improves readability without sacrificing precision, and we have adopted the proposed wording. We also used this comment as a cue to scan the manuscript for similarly dense expressions and adjusted them where appropriate (page 2, lines 58-60).
Comment 13: Some references have inconsistent punctuation and italicization. Example: Entry 3: "Bibikova, M.; Fan, J.-B. Genome-wide DNA methylation profiling. WIREs Syst. Biol. Med. 2010, 2, 210–223. https://doi.org/10.1002/wsbm.35." → Ensure journal names are italicized consistently: "WIREs Syst. Biol. Med." → "WIREs Syst. Biol. Med." (correct, but verify consistency with MDPI guidelines). Duplicate reference (Entry 18) should be removed.
Response: We thank the reviewer for carefully checking the reference list. We agree that consistent formatting and removal of duplicates are essential.
Reference formatting and journal italicization: We have carefully reviewed all references to ensure that journal titles are consistently italicized and that punctuation and spacing conform to MDPI style (e.g., use of commas, semicolons, and DOI formatting). Specifically, Entry 3 has been checked and now appears in MDPI format as:
Bibikova, M.; Fan, J.-B. Genome-Wide DNA Methylation Profiling. WIREs Syst. Biol. Med. 2010, 2, 210–223. https://doi.org/10.1002/wsbm.35.
Duplicate reference (Entry 18) and substitution: The duplicate Reference 18 has been removed.
To maintain the numbering sequence without renumbering all subsequent citations, we have replaced Reference 18 with a different, relevant review article on bacterial epigenetic regulation:
- Adhikari, S.; Curtis, P.D. DNA Methyltransferases and Epigenetic Regulation in Bacteria. FEMS Microbiol. Rev. 2016, 40, 575–591. https://doi.org/10.1093/femsre/fuw023.
Comment 14: Table 1’s header "Functional Therapeutic Target/Strategy" is ambiguous. Split into two columns: "Functional Role" and "Therapeutic Target/Strategy" to distinguish between the biological function of the epigenetic mechanism and its therapeutic application. Table 2’s "Example Compound/Tool" column has inconsistent formatting (e.g., "dCas9-TET1, dCas9-Dam" vs. "Ribostatin")—use consistent punctuation (e.g., commas for multiple tools, no extra spaces).
Response: We thank the reviewer for these precise and helpful suggestions regarding table clarity and formatting. We agree that separating biological function from therapeutic application in Table 1 and standardizing the formatting of examples in Table 2 improves readability and interpretability.
Comment 15: Each section/subsection should open with a clear topic sentence that summarizes the core idea. Example: Section 2.3 ("RNA Methylation and Small Regulatory RNAs") currently opens with "Beyond DNA, RNA methylation fine-tunes translation and antibiotic target accessibility." Revise to: "Beyond DNA methylation, RNA-level epigenetic regulation—including RNA methylation and small non-coding RNAs (sRNAs)—fine-tunes translation and antibiotic target accessibility, contributing to adaptive resistance." This clarifies the section’s scope upfront.
Response: We thank the reviewer for this obvious and helpful suggestion. We agree that explicit topic sentences at the start of each subsection improve readability and help orient the reader to the scope and purpose of the section. We have adopted the reviewer’s proposed revision to Section 2.3 and have also reviewed other subsections to ensure their opening sentences clearly summarize the core idea (page 4, lines 183-185).
We sincerely thank Reviewer 2 for the careful, in-depth evaluation of our manuscript and for the many thoughtful questions and detailed language suggestions. Your comments prompted us to clarify the distinction between correlative and causal epigenetic evidence, better frame cross-kingdom “universality,” temper translational claims (especially for CRISPR-based tools and epidrugs), and refine our discussion of AI models and microbiome-wide interactions. We have also implemented your recommendations on structure, section transitions, figure legends, tables, abbreviations, and wording to improve readability and consistency. We believe these revisions have substantially strengthened both the scientific rigor and clarity of the manuscript, and we are very grateful for the time and expertise you devoted to this review.
Round 2
Reviewer 1 Report
Comments and Suggestions for Authors
This has been well revised. There are few minor comments.
- 1 – many typos and actually figures do not quite match to the contents. Carefully check the words and images in the illustrations.
A: “Bacterial DNA methylation regulates efflux genes and promotes their expression.”
B: i) “Fungal chromatin modifications via histone acetyl- and methyltransferases promote stress resistance.” ii) correct spelling out the “HAT”, iii) figure does not make sense. (Me, Me)
C: delete one “Adaptation” and “RNA-based regulation via m⁶A modifications influencing translation”
D: “Cross-kingdom conservation of methylation and acetylation networks coordinating microbial epigenetic plasticity”
- The same comments for other figures as Fig. 1.
- Tables – revise mismatched contents in the Tables. And, most references are not quite matched. These seems to be simply generated by AI tools.
Author Response
We sincerely thank the reviewer for the positive overall evaluation of the revised manuscript and for the additional minor comments. We appreciate the time and care invested in this second round of review. We have carefully addressed each point below and revised the figures and tables accordingly. All changes are incorporated in the revised manuscript, and the corresponding locations (figures/tables and sections) are indicated below.
Comment 1
- 1 – many typos and actually figures do not quite match to the contents. Carefully check the words and images in the illustrations.
A: “Bacterial DNA methylation regulates efflux genes and promotes their expression.”
B: i) “Fungal chromatin modifications via histone acetyl- and methyltransferases promote stress resistance.” ii) correct spelling out the “HAT”, iii) figure does not make sense. (Me, Me)
C: delete one “Adaptation” and “RNA-based regulation via m⁶A modifications influencing translation”
D: “Cross-kingdom conservation of methylation and acetylation networks coordinating microbial epigenetic plasticity”
Response: We thank the reviewer for these detailed and constructive comments on Figure 1. We carefully reviewed the figure for typos, wording, and alignment between the text and the graphical elements. We have now revised the labels and layout of all panels (A–D) and corrected the figure legend to ensure that the wording accurately matches the illustrations and the main text (page 7).
Comment 2
- The same comments for other figures as Fig. 1.
Response: We agree that the same level of care should be applied to all figures. In response, we systematically re-checked each figure (Figure 2) for typos, label consistency, and alignment between the illustrations and the textual descriptions (page 11). The same was applied for Figure 3 (page 15) and Figure 4 (page 22).
Comment 3
Tables – revise mismatched contents in the Tables. And, most references are not quite matched. These seems to be simply generated by AI tools.
Response: We thank the reviewer for highlighting this critical issue and apologize for the earlier inconsistencies. In the revised manuscript, we have rechecked and corrected all tables: Table 1 (pages 6-7), Table 2 (pages 15-17), and Table 3 (page 22).
We would like to sincerely thank the reviewer once again for their careful re-evaluation of our manuscript and for the constructive minor comments provided in this second round. Your detailed suggestions on the figures, tables, and references, as well as the precision of wording, have helped us to correct remaining inconsistencies and further polish the manuscript. These final revisions have improved the clarity, accuracy, and overall presentation of our work, and we are very grateful for the time and expertise you have devoted to this review process.